https://doi.org/10.1038/s42003-022-04275-y | OPEN
# Gamete dimorphism of the isogamous green alga (*Chlamydomonas reinhardtii*), is regulated by the mating type-determining gene, *MID*

Ryoya Innami[1], Shinichi Miyamura [1✉], Masako Okoshi[1], Tamotsu Nagumo[2], Kensuke Ichihara[3], Tomokazu Yamazaki[4] & Shigeyuki Kawano[4]

The gametes of chlorophytes differ morphologically even in isogamy and are divided into two types (α and β) based on the mating type- or sex-specific asymmetric positioning of the mating structure (cell fusion apparatus) with respect to the flagellar beat plane and eyespot, irrespective of the difference in gamete size. However, the relationship between this morphological trait and the mating type or sex determination system is unclear. Using mating type-reversed strains of the isogamous alga *Chlamydomonas reinhardtii*, produced by deletion or introduction of the mating type-determining gene *MID*, we revealed that the positioning of the mating structure is associated with conversion of mating types (mt⁻ and mt⁺), implying that this trait is regulated by *MID*. Moreover, the dominant mating type is associated with the type β phenotype, as in the chlorophyte species *Ulva prolifera*. Our findings may provide a genetic basis for mating type- or sex-specific asymmetric positioning of the chlorophyte mating structure.

[1] Faculty of Life and Environmental Sciences, University of Tsukuba, Tsukuba, Ibaraki 305-8572, Japan. [2] Echigo Natural History Laboratory, Ojiya, Niigata 947-0041, Japan. [3] Field Science Center for Northern Biosphere, Hokkaido University, Funami-cho, Muroran 051-0013, Japan. [4] Graduate School of Frontier Sciences, The University of Tokyo, Wakashiba, Kashiwa, Chiba 277-0871, Japan. ✉email: miyamura.shinichi.fw@u.tsukuba.ac.jp

Two sexes (male and female) are readily distinguishable based on differences in gamete size in anisogamous and oogamous multicellular eukaryotes[1,2]. Males produce small gametes or sperms, whereas females produce large gametes or egg cells (gamete size dimorphism). In contrast, ancestral isogamous species, from which anisogamy and oogamy are almost certainly derived, are usually found in single-celled eukaryotes and have two or more mating types.

The gametes of isogamous species are of similar size and appearance. Therefore, two gametes belonging to opposite mating types generally cannot be distinguished from each other based on their size or morphology in isogamous species. Such features of the gametes are crucial because they define mating type and sex[2]. Even so, bipolar sexual differentiation is present in isogamous species, e.g. cytoplasmic inheritance, gamete recognition and adhesion mechanisms and prefusion mating behaviour[3], and so these dimorphisms must have preceded the evolution of two sexes.

Studies of chlorophyte algae indicate that two gametes of opposite mating types can be distinguished based on their morphology, irrespective of the gamete size difference[4–6]. This morphological feature is a mating type- or sex-specific asymmetric positioning of the mating structure (cell fusion apparatus of green algae) and/or cell fusion site of the gamete, which occupies different positions between the opposite mating types or sexes (Fig. 1). The gamete can be divided into two morphological types (α and β) based on this difference. This trait is a type of sexual dimorphism at the level of gamete structure (gamete dimorphism) and was discovered in the volvocine green alga *Chlamydomonas reinhardtii* by light microscopy[4] and was confirmed by electron microscopy[7,8]. *C. reinhardtii* is a flagellate alga with two mating types, mating type plus (mt[+]) and mating type minus (mt[−]), and is isogamous. The cell has two flagella elongated from the basal bodies, each with special microtubules termed microtubular roots, and one eyespot (photoreceptive apparatus), which consists of a red carotenoid pigmented area in a chloroplast and photoreceptor in an overlying region of the plasma membrane (Fig. 1a, b). The eyespot is associated with one of the four microtubular roots and located near the cell equator. Each gamete normally contains a single eyespot and mating structure arranged asymmetrically around the anterior–posterior axis of the cell (Fig. 1c)[4]. The mating structure of mt[+] gametes is located at the cell apex on the side of the flagellar beat plane opposite the eyespot (*anti* side, according to the imaginary *syn/anti* plane bisecting the cell proposed by Holmes and Dutcher[4]) (type α), whereas the mt[−] structure is located on the same side as the eyespot (*syn* side) (type β) (Fig. 1c)[4,9]. We use the term 'mating structure position' (MSP) type α and β for the former and the latter arrangements, respectively, and defined gametes with MSP of type α and β as type α and β gametes, respectively. Such asymmetric arrangement of the mating structure is presumably determined by the microtubular roots[10]. The mt[+] structure is associated with the 2d root, and the mt[−] structure is associated with the 1d root[4,7], according to the numbering system for basal

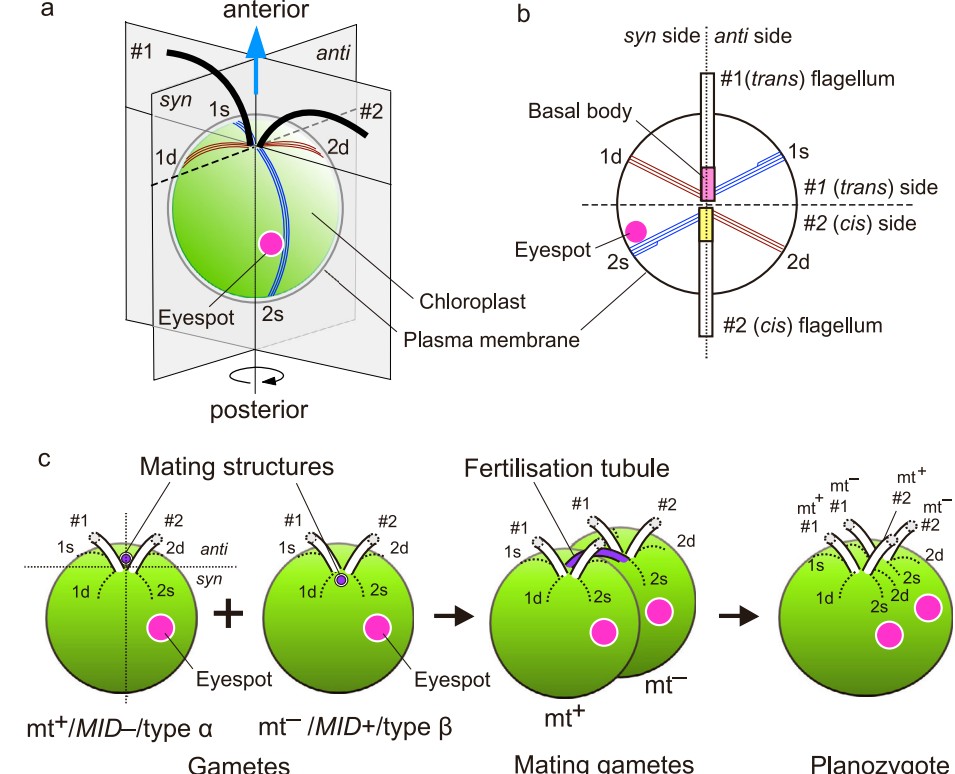

**Fig. 1 Schematic diagram of the spatial arrangement of flagellum–eyespot–mating structures in the vegetative cell, gametes, mating gametes and planozygote of *Chlamydomonas reinhardtii*. a** Three-dimensional image of the vegetative cell. The cell wall and probasal bodies are not depicted in the diagram. The blue arrow indicates the direction and axis of cell movement. **b** Asymmetric organisation of the cytoskeleton and eyespot viewed from the cell anterior in the vegetative cell. **c** Gamete fusion and planozygote formation. The mating structure of the mt[+] gamete is located on the side of the beat plane opposite the eyespot (type α), whereas that of the mt[−] gamete is located on the same side as the eyespot (type β). Gamete fusion occurs between a fertilisation tubule elongated from an mt[+] mating structure and mt[−] structures. Next, the cell fusion extends between the area circumscribed by 1s–2d microtubular roots in mt[+] gametes and 1d–2s roots in mt[−] gametes. In the planozygote, two flagellar pairs align in parallel, and two eyespots are positioned side by side on the same side of the cell. anti: *anti* side, syn: *syn* side, #1: no. 1 flagella, #2: no. 2 flagella, *1d, 1s, 2d, 2s*: 1d, 1s, 2d, 2s microtubular roots, respectively (**a, b**), or lateral ridges overlying 1d, 1s, 2d, 2s roots, respectively (**c**), *MID*+: presence of *MID*, *MID*−: absence of *MID*.

bodies (no. 1 for the older basal body; no. 2 for the younger basal body) and microtubular roots (1d and 1s roots attached to the no. 1 basal body; 2d and 2s roots attached to the no. 2 basal body)[11]. As a result of mating type- or sex-specific asymmetric arrangement of the mating structures, a swimming zygote (planozygote) with parallel flagellar pairs and two eyespots on the same side of the cell forms after gamete fusion (Fig. 1c).

In contrast to other morphological features specific to one of the two mating types or sexes, such as the $mt^+$-specific elongation of a fertilisation tubule from the mating structure of *C. reinhardtii*[12], which is found only in close relatives of this species[13], asymmetric positioning of the mating structure/cell fusion sites is prevalent in many iso- and anisogamous species and is likely a universal feature of chlorophytes[5,6,14–17]. In addition, this trait may be present in other eukaryotes; a similar phenomenon has been reported in the mating gametes of a dinoflagellate (Alveolata)[18,19], although its mating type specificity has not been determined. Therefore, this trait can provide insights into the origin and evolution of gamete dimorphism not only in chlorophytes but also in other eukaryotes, and is independent of other differences between mating types in isogamy[3] and gamete size dimorphism in anisogamy[20]. However, the genetic background of this trait needs to be investigated.

Ulvophycean algae provide insights into the genetic background of the asymmetric positioning of the mating structure (type α and β MSPs). In the anisogamous species of ulvophycean algae examined to date, male and female gametes are type α and β gametes, respectively, implying that the positioning of the mating structure and/or cell fusion site may be closely related to a particular sex or mating type[9]. Mating type-specific genes were found in the mating type locus of the green seaweed *Ulva partita*[21]. Also, the MSP (α and β) is correlated with the presence or absence of mating type-specific genes in this and the closely related species *U. prolifera*[21,22], implying that the positioning of the mating structure is a mating type- or sex-specific trait regulated by the mating type locus or sex-determining gene. However, this relationship has not been empirically examined because the mating type- or sex-determining gene has not been identified in *Ulva* or other ulvophycean species. Therefore, whether the MSP is directly regulated by the mating type- or sex-determining gene is unclear.

We investigated the causal relationship between the spatial positioning of the gamete mating structure and the mating type- or sex-determining gene. For this purpose, we used *C. reinhardtii* because the mating type- or sex-determining gene has been identified in this species and other volvocine species[23–26] but not in other chlorophyte algae. The mating type of *C. reinhardtii* is regulated by the mating type-determining gene *MID* (minus dominance), which is located in the R domain of the mating type locus of $mt^-$[23,27]. The cell differentiates to $mt^-$ in the presence of *MID* and to $mt^+$ in its absence. *MID* is a dominant determinant of $mt^-$ and encodes a putative RWP-RK family transcription factor that activates the genes involved in functions specific to $mt^-$ and represses those involved in functions specific to $mt^+$[23]. Consequently, $mt^-$ and $mt^+$ cells produce the proteins required for gametogenesis and fertilisation in gametes of each mating type. Nevertheless, the gene(s) involved in the positioning of the mating structures is obscure, and it is unknown whether its positioning is directly regulated by *MID*. We observed the positioning of the mating structure of the wild-type and mating type-reversed strains (from $mt^-$ to $mt^+$ and vice versa), which were produced through deletion or introduction of *MID*[27,28], using light, fluorescence, and field emission scanning electron microscopy (FE-SEM). We demonstrated that the spatial positioning of the gamete mating structure was replaced in association with the reversion of mating type from $mt^-$ to $mt^+$ and vice versa,

indicating that the positioning of the mating structure is regulated by *MID*. We also found that the dominant mating type was associated with the type β phenotype using a heterozygous diploid gamete. Finally, we discussed the importance of the tight association of *MID* with asymmetric positioning of the mating structure for proper phototactic behaviour of *C. reinhardtii* and the possibility of genetic control of this trait by mating type (sex)-determining genes in *Ulva* and other chlorophytes.

## Results

**Positioning of the mating structure is replaced in association with the reversion of mating type from $mt^-$ to $mt^+$.** To examine the role of the mating type-determining gene *MID* in the positioning of the gamete mating structure, we used the wild-type (CC-125 ($mt^+$) and CC-124 ($mt^-$)) and mating type-reversed strains (CC-3712 (mid $mt^-$) and CC-3947 ($mt^+$ T-MID)), which are produced by deletion or introduction of *MID*, respectively (Supplementary Fig. 1)[27,28]. Two eyespots align on the same side of the planozygote after fertilisation of wild-type gametes because the position of the mating structure is different between $mt^+$ and $mt^-$ gametes of the wild-type strains[4] (Fig. 1c). Therefore, it is possible to use this feature to estimate the positioning of the mating structure in the gametes of mating type-reversed strains. However, CC-3712 (mid $mt^-$) gametes could produce fertilisation tubules but did not fuse with $mt^-$ or $mt^+$ gametes because this strain lacks the cell-adhesion gene *FUS1*[29] (Supplementary Fig. 1). Therefore, we directly observed fertilisation tubule elongation from the $mt^+$ mating structure. The outgrowth of a fertilisation tubule of wild-type ($mt^+$) and CC-3712 gametes was induced by 10 mM dibutyryl-cAMP and 1 mM 3-isobutyl-1-methylxanthine and verified by immunofluorescence microscopy using an anti-actin antibody (Fig. 2a–d) because the fertilisation tubule is composed of actin filaments[30]. In the wild-type ($mt^+$) and CC-3712 gametes, the fertilisation tubule elongated from the cell apex (Fig. 2a, c). Subsequently, the gametes were observed by FE-SEM to determine the precise spatial position of the fertilisation tubule. In the wild-type gamete, the fertilisation tubule was present on the side of the beat plane opposite the eyespot (Fig. 2e) and most frequently at the intersection point of the two lateral ridges overlying the 1s and 2d roots (Fig. 2f), as reported previously in IAM C-541 (= NIES-2238) ($mt^+$)[8]. Occasionally, the fertilisation tubule was present on the lateral ridges corresponding to the 1s or 2d roots. In CC-3712, the fertilisation tubule occupied the same position as that of the wild-type ($mt^+$) gametes (Fig. 2g–i).

**Positioning of the fertilisation tubule with respect to microtubular roots.** To confirm these results, we observed the position of the fertilisation tubule with respect to the flagellar beat plane and four microtubular roots (1d, 1s, 2d and 2s) of the gamete by fluorescence microscopy using anti-acetylated tubulin and -actin antibodies. Figure 3a shows fluorescence micrographs of activated wild-type ($mt^+$) and CC-3712 gametes. In both gametes, the flagella and microtubular roots were stained with an anti-acetylated tubulin antibody. We identified the four roots based on their acetylation and length after staining with an anti-acetylated tubulin antibody using the 2s root as a positional marker, as this root is the most extensively acetylated or longest of the four roots and is associated with the eyespot in *C. reinhardtii*[31]. The lengths of the four microtubular roots of wild-type ($mt^+$) and CC-3712 are shown in Table 1. In almost all cells, one microtubular root was the longest ($4.18 \pm 1.09$ μm [mean ± SD] in the wild-type ($mt^+$) and $4.17 \pm 0.85$ μm [mean ± SD] in CC-3712) compared with the other roots (Tukey's test, $p < 0.01$) and was identified as the 2s root. Next, we analysed the position of the fertilisation tubule, which was labelled with an anti-actin antibody, with respect to

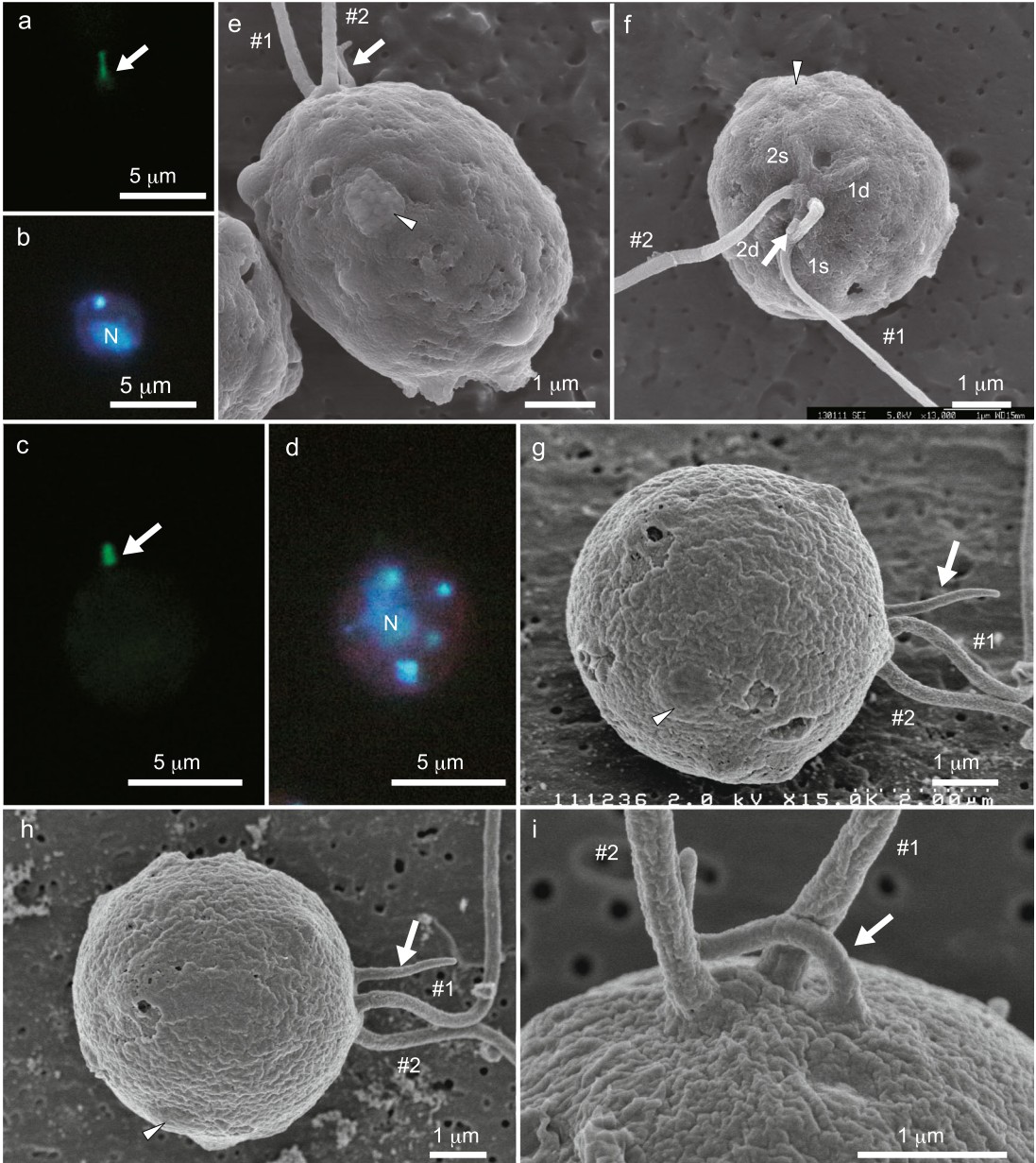

**Fig. 2 Localisation of the fertilisation tubule and eyespot in wild-type CC-125 (mt⁺) and mating type-reversed CC-3712 (mid mt⁻) gametes.** The fertilisation tubule and nucleus were reacted with an anti-actin antibody (**a**, **c**) and DAPI (**b**, **d**), respectively. Spatial positioning of the fertilisation tubules and eyespots was visualised by FE-SEM (**e–i**). **a**, **b** Wild-type mt⁺ gamete. The fertilisation tubule was elongated from the cell apex. **c**, **d** CC-3712 gamete. The fertilisation tubule was extended from the cell apex. **e** Side view of the wild-type mt⁺ gamete. The fertilisation tubule was present on the side of the beat plane opposite the eyespot. **f** Top view of the wild-type mt⁺ gamete. The fertilisation tubule was situated on the intersection point of the two lateral ridges overlying the 1s and 2d roots. **g** Side view of the CC-3712 gamete. **h** Tilted image of **g**. The fertilisation tubule was present on the side of the beat plane opposite the eyespot. **i** Enlarged image of the cell anterior of a CC-3712 gamete. The fertilisation tubule was elongated from the flagellar base close to the #1 flagellum. Arrows and arrowheads indicate fertilisation tubules and eyespots, respectively. #1: no. 1 flagellum, #2: no. 2 flagellum, *1d, 1s, 2d* and *2s*: lateral ridges overlying 1d, 1s, 2d and 2s microtubular roots, respectively, *N*: nucleus.

the beat plane and the microtubular roots, using the 2s root as a positional marker. In both strains, the fertilisation tubule was almost always localised on the *anti* side of the beat plane and was typically present at the intersection point of the 1s and 2d roots (77.4% in the wild-type and 68.5% in CC-3712) or 1s root (17.9% in the wild-type and 28.6% in CC-3712) (Tukey's test, $p < 0.01$) (Fig. 3b, c). Regarding the fertilisation tubule distribution pattern, the chi-squared test showed no significant difference between the two strains ($p = 0.1 > 0.05$). Taken together, these results imply that the fertilisation tubule of CC-3712 gametes is always situated

on the side of the beat plane opposite the 2s root, which is always associated with the eyespot.

**Positioning of the mating structure is replaced in association with the reversion of mating type from mt⁺ to mt⁻.** For CC-3947 (mt⁺ *T-MID*), we indirectly examined the position of the mating structure in mating gametes by FE-SEM, using the fertilisation tubule and eyespot as positional markers, because it was difficult to visualise the mt⁻ mating structure directly. Figure 4a

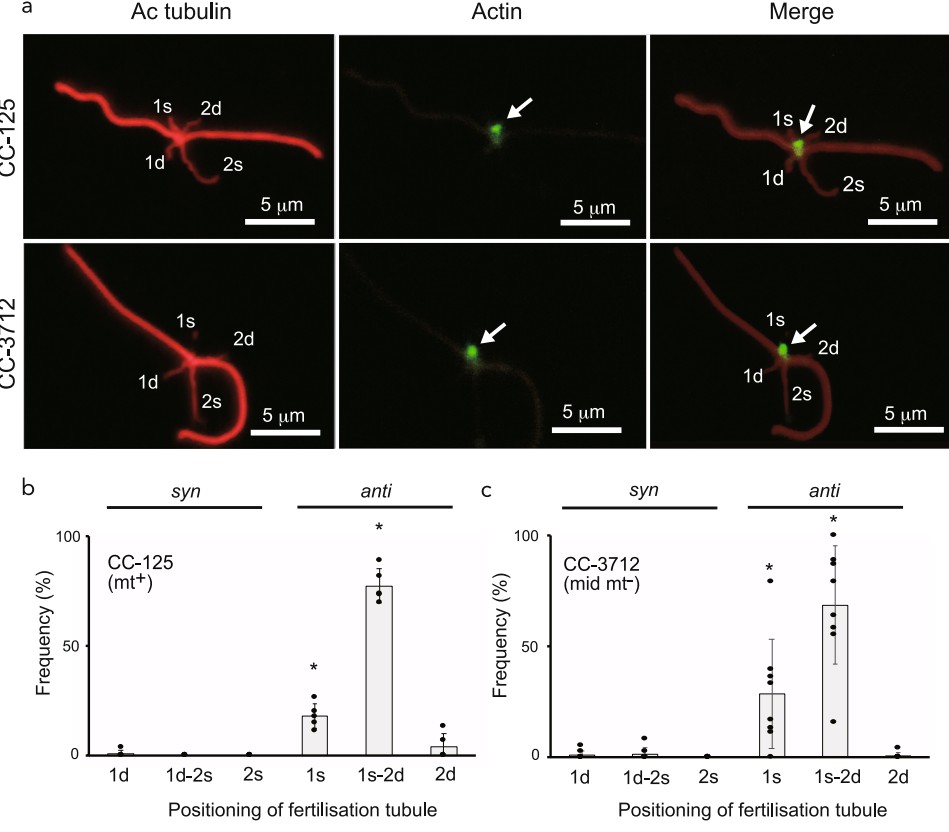

**Fig. 3 Positioning of the fertilisation tube with respect to the flagellar beat plane and four microtubular roots in the wild-type CC-125 (mt+) and mating type-reversed strain CC-3712 (mid mt⁻).** **a** Immunofluorescence staining of wild-type CC-125 (mt+) and mating type-reversed strain CC-3712 (mid mt⁻) gametes. Cells were stained with an anti-acetylated tubulin antibody (Ac tubulin) and an anti-actin antibody. Arrows indicate fertilisation tubules. **b, c** Frequency of the positioning of the fertilisation tubule in the wild-type mt+ (**b**) and mating type-reversed strain CC-3712 (**c**). Values are means of five (CC-125) and eight (CC-3712) independent observations ($n = 15, 18, 19, 23$ and 27 in CC-125; $n = 9, 12, 18, 19, 22, 23, 24$, and 38 in CC-3712). Bars indicate standard deviations. *$p < 0.01$ by Tukey's test. *1d, 1s, 2d*, and *2s*: 1d, 1s, 2d and 2s microtubular roots, respectively.

**Table 1 Lengths of the four microtubular roots of gametes.**

| Strain | Length of microtubular root (µm) | | | |
|---|---|---|---|---|
| | 1d | 1s | 2d | 2s |
| CC-125 | 2.16 ± 0.62 | 2.02 ± 0.53 | 2.02 ± 0.62 | 4.18 ± 1.09* |
| CC-3712 | 2.17 ± 0.43 | 2.02 ± 0.45 | 2.16 ± 0.46 | 4.17 ± 0.85* |

Values are presented as mean ± standard deviation. 2s length significantly differed (*$p < 0.01$ with Tukey's test, $n = 100$) from the lengths of other roots in each strain.

shows the wild-type (mt+) × CC-3947 gamete pair 15 min after mixing of the gametes, in which two gametes joined by a fertilisation tubule elongated from a wild-type (mt+) mating structure. The base of the fertilisation tubule was on the side of the beat plane opposite the eyespot in the left cell; therefore, the left cell is a wild-type (mt+) gamete. In the right cell, the fertilisation tubule was attached to the base of two flagella and occupied the same side as the eyespot with respect to the beat plane. Therefore, the right cell is a CC-3947 gamete. These results imply that the mating structure of CC-3947 is present on the same side of the flagellar beat plane as the eyespot, similar to wild-type (mt⁻) gametes. Furthermore, we used the position of the planozygote eyespots to evaluate the position of the mating structure in the gametes and observed the alignment of the eyespots in the planozygotes using light microscopy. For this purpose, we used the wild-type (mt+) strain because the positions of the mating structure and eyespot are established (Fig. 2e, f). Figure 4b, c

shows the planozygote 15 min after mixing wild-type (mt+) and CC-3947 gametes, in which two eyespots aligned on the same side of the cell in the same way as in the wild-type (mt+) × wild-type (mt⁻) pair (Fig. 4d, e). One eyespot occupied a position close to the cell fusion plane, whereas the other eyespot was far from the plane. Such alignment of two eyespots on the same side of the cell was found in 145 and 144 of 150 planozygotes in wild-type (mt+) × CC-3947 (mt+ T-MID) and in wild-type (mt+) × wild-type (mt⁻) pairs, respectively (binomial test, $p < 0.01$, parameter $\theta = 0.5$) (Fig. 4f). Because the mating structure of the wild-type (mt+) gametes is on the side of the flagellar beat plane opposite the eyespot, the left part of the planozygote in Fig. 4b, c was derived from a wild-type (mt+) gamete and the right part from a CC-3947 gamete, indicating that the mating structure of the CC-3947 gamete is on the same side as the eyespot.

**Type β phenotype is associated with dominant mating type.** Next, we examined which MSP (α or β) is preferentially expressed in a heterozygous diploid strain (CC-127) harbouring both mt+- and mt⁻-specific genes (Supplementary Fig. 2) and behaving as mt⁻. If the spatial arrangement of the mating structure is a mating type-specific trait regulated by *MID*, heterozygous diploid gametes would exhibit a type β phenotype, similar to other mt⁻-specific traits, because mt⁻ is the dominant mating type regulated by *MID*[23] in cells heterozygous for mating type[32]. We used the position of the planozygote eyespots to evaluate the position of the mating structure in the diploid gametes. In contrast to wild-type (mt+) gametes (Fig. 5a), CC-127 diploid gametes

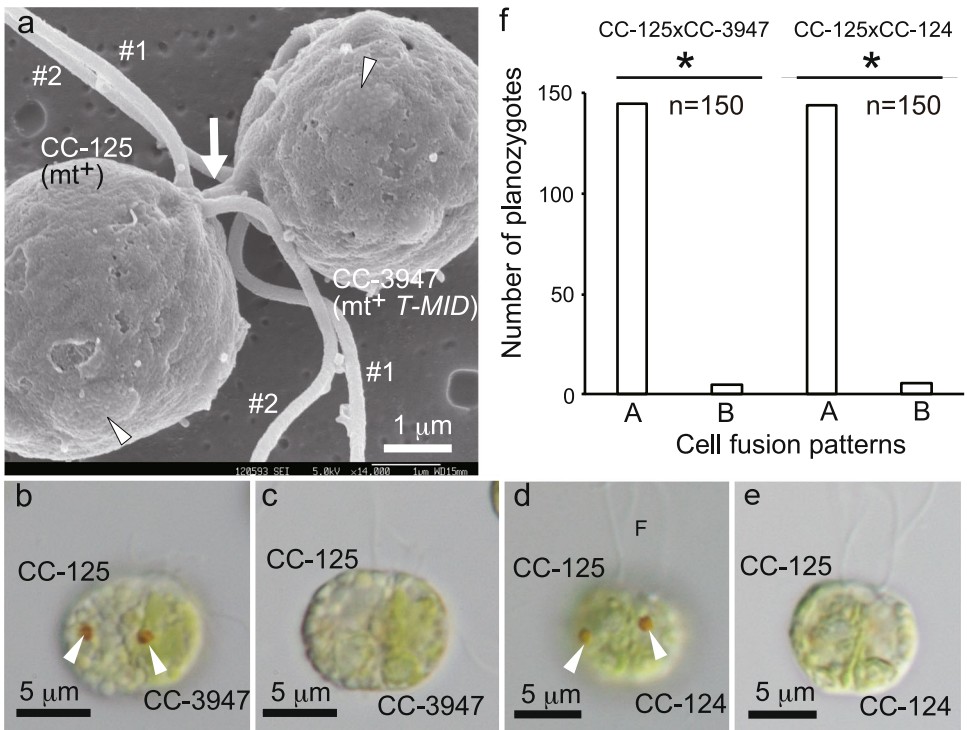

**Fig. 4 Spatial positioning of the mating structure in the mating type-reversed strain CC-3947 (mt⁺ *T-MID*) gamete. a** Positioning of the mating structure of the CC-3947 gamete observed by FE-SEM using mating gametes 15 min after mixing wild-type CC-125 (mt⁺) and CC-3947 gametes. The fertilisation tubule elongated from the mt⁺ mating structure fused to that of CC-3947, which was present on the same side of the beat plane as the eyespot. Arrow and arrowheads indicate the fertilisation tubule and eyespots, respectively. **b, c** Differential interference contrast images of the planozygote 15 min after mixing wild-type CC-125 (mt⁺) and CC-3947 gametes. **d, e** Differential interference contrast images of the planozygote at 15 min after mixing wild-type CC-125 (mt⁺) and CC-124 (mt⁻) gametes. Images were focused on the upper (**b, d**) and lower (**c, e**) surface of the cell, respectively. **f** Number of planozygotes with two eyespots on the same side (A) or opposite side (B) of the cell at 15–30 min after mixing the gametes. Similar results were obtained from three independent experiments (*n* = 150, 126, and 133 in CC-125 × CC-3947; *n* = 150, 358, and 465 in CC-125 × CC-124), and a representative result is shown. *Significant difference between (A) and (B) by the binomial test (*p* < 0.01, parameter *θ* = 0.5). #1: no. 1 flagellum, #2: no. 2 flagellum, F: flagellum.

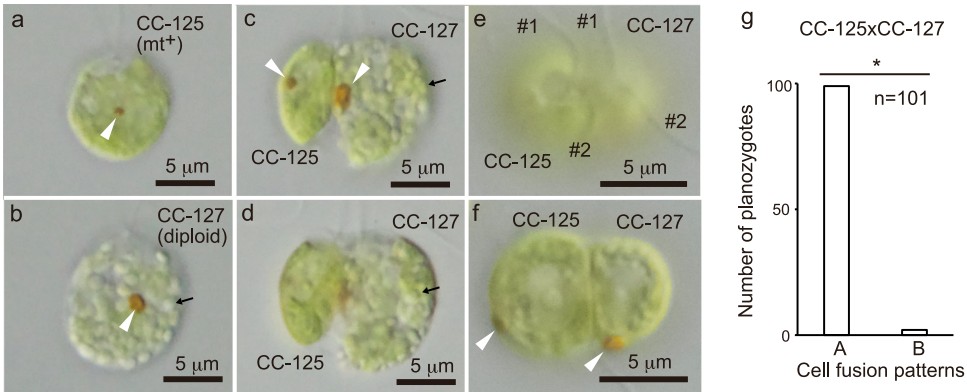

**Fig. 5 Spatial positioning of the mating structure in heterozygous diploid strain CC-127 gamete. a, b** Differential interference contrast images of the wild-type CC-125 (mt⁺) (**a**) and CC-127 (mt⁻) gametes (**b**). In contrast to CC-125, the CC-127 gamete was filled with starch grains (arrow). **c, d** Differential interference contrast side-view images of the planozygote 25 min after mixing wild-type CC-125 (mt⁺) and CC-127 (mt⁻) gametes. Two eyespots aligned on the same side of the cell. **e, f** Differential interference contrast top-view images of the planozygote 25 min after mixing wild-type CC-125 (mt⁺) and CC-127 (mt⁻) gametes. Gamete–gamete fusion occurred on the side of the beat plane opposite the eyespot in the wild-type (mt⁺) gamete and on the same side in the diploid gamete. Images were focused on the upper surface (**c, e**), lower surface (**d**) and equator (**f**) of the cell body, respectively. Arrowheads indicate the eyespots. **g** Number of planozygotes with two eyespots on the same side (A) or opposite side (B) of the cell at 25 min after mixing the gametes. Similar results were obtained from three independent experiments (*n* = 101, 102 and 103), and a representative result is shown. *Significant difference between (A) and (B) by the binomial test (*p* < 0.01, parameter *θ* = 0.5). #1: no. 1 flagellum, #2: no. 2 flagellum.

accumulated starch grains (Fig. 5b), enabling distinction of CC-127 gametes from wild-type (mt$^+$) gametes. Figure 5c, d shows a side view of the planozygote 25 min after mixing wild-type (mt$^+$) and diploid gametes, in which two eyespots aligned on the same side of the cell. Figure 5e, f shows a top view of the planozygote, in which gamete–gamete fusion occurred on the side of the beat plane opposite the eyespot in wild-type (mt$^+$) gametes and on the same side of the beat plane as the eyespot in diploid gametes. Such alignment of two eyespots on the same side of the cell was found in 99 of 101 planozygotes (binomial test, $p < 0.01$, parameter $\theta = 0.5$) (Fig. 5g). These results imply that the type β phenotype is preferentially expressed in heterozygous diploid gametes and is associated with the dominant mating type.

## Discussion

Using mating type-reversed strains of *C. reinhardtii*, we revealed that the spatial positioning of the gamete mating structure was replaced in association with the reversion of mating type from mt$^-$ to mt$^+$ and vice versa. The mating structure was located on the side of the beat plane opposite the eyespot (type α) in the wild-type (mt$^+$) and CC-3712 (mid mt$^-$) versus the same side of the eyespot (type β) in wild-type (mt$^-$) and CC-3947 (mt$^+$ T-MID). CC-3712 was originally mt$^-$, but it lacks *MID* and thus behaves as pseudo-plus[27], implying that the MSP changed from type β to α in association with deletion of *MID* and reversion of the mating type from mt$^-$ to mt$^+$. CC-3947 was originally mt$^+$ but retains the *MID* transgene and mates as mt$^-$[28], implying that the MSP changed from type α to β in association with introduction of *MID* into the mt$^+$ genome and reversion of the mating type from mt$^+$ to mt$^-$. However, in CC-3947, it is possible that insertion of *MID* at a different site from the original position affects the spatial positioning of the mating structure irrespective of the reversion of mating type; however, this is unlikely because the heterozygous diploid strain CC-127, which has *MID* in the mt$^-$ chromosome and differentiates as a mt$^-$ gamete, also exhibited a type β phenotype. Both CC-127 and CC-3947 carry *MID*, behave as mt$^-$ gametes, and exhibit the type β phenotype, despite their different genetic backgrounds. Therefore, the reversion of MSP from α to β can be attributed to the effect of the *MID* transgene in CC-3947. Consequently, our results imply that the spatial positioning of the mating structure is regulated directly by *MID*, as are other mating type-specific traits.

The positioning of the mating structure is determined by the microtubular root[10,33]. The mt$^+$ structure is associated with the 2d root and the mt$^-$ structure with the 1d root[7,34]. Immuno-fluorescence observation of wild-type (mt$^+$) and CC-3712 gametes revealed that the fertilisation tubule/mt$^+$ mating structure is preferentially associated with the 1s-2d, 1s, or 2d microtubular root, in that order. This is consistent with a previous study using FE-SEM[8], in which the fertilisation tubule elongated from the intersection point of 1s and 2d roots. By contrast, Goodenough and Weiss[10] and Gaffal et al.[7], reported that the mt$^+$ mating structure is associated with the d/2d root, and Weiss[35] reported that the mt$^+$ mating structure is associated with the 3-over-1 root (s root). It is unclear why the position of the fertilisation tubule/mt$^+$ mating structure is different in these studies. One possibility is that the position of the mt$^+$ mating structure varies among strains or laboratory cultures derived from the same strain, as this and the previous studies used the same wild-type strains (CC-125 (137c) in this study, 137c in Goodenough and Weiss[10], and C-541 (137c) in Miyamura et al.[8]), while other research involved a different strain (RC3[35]). Nevertheless, all of the studies indicated that the mt$^+$ mating structure is associated with the microtubular root on the side of the eyespot opposite the beat plane. To organise the mating structure through

the microtubular root, *MID* probably regulates the expression of the protein(s) that directly connects the mating structure to the microtubular root or system I fibre associated with the micro-tubular root[33]. This protein(s) probably connects the mating structure to the 1d roots in the presence of *MID* but to the 1s and/or 2d roots in the absence of *MID*. To verify this hypothesis, it is necessary to identify the genes downstream of *MID* involved in the mating structure–microtubular root association or determination of the *anti*/*syn* side of the cell.

A tight association between *MID* and the spatial positioning of the mating structure may be crucial for survival of the planozygote of *C. reinhardtii*, which exhibits negative phototaxis[36] and swims briefly before settling and forming a resistant thick-walled cyst (zygospore) in freshwater and soil habitats[13]. During this process, four flagella and two eyespots of the planozygote probably play a crucial role in phototaxis. The cell rotates around its axis while swimming and finds the direction of a light source by scanning the environment using a photoreceptor in the plasma membrane and underlying reflective carotenoid-pigmented area of an eyespot[37]. The light signal detected by the photoreceptor is transmitted to the flagella as an electron signal, leading to influx of Ca$^{2+}$ into the flagella[38]. The sensitivity to Ca$^{2+}$ differs between the two flagella (nos. 1 and 2) in *C. reinhardtii*[39], such that the cell can turn toward or away from the light source by controlling the flagellar beat balance. For quadriflagellate planozygotes, two no. 1 flagella beat in the same direction after receiving the light signal from each eyespot, as do two no. 2 flagella. Considering such features, Holmes and Dutcher[4] proposed that planozygotes with parallel flagellar pairs (two no. 1 and two no. 2 flagellar pairs) and two eyespots on the same side of the cell may be required for proper phototactic behaviour, and the mating type-specific asymmetric positioning of the mating structure (Fig. 6a) has evolved in *C. reinhardtii* to ensure such an arrangement of flagella and eyespots.

However, there are at least three alternative possibilities (Fig. 6b–d). In the first (Fig. 6b) and second (Fig. 6c) cases, both mt$^+$ and mt$^-$ gametes have the mating structure on the same side (Fig. 6b) or opposite side (Fig. 6c) of the beat plane as the eyespot. In these cases, each eyespot is placed on the opposite side of the cell, and no. 1 and no. 2 flagella form a pair in the planozygote (Fig. 6b, c). Such planozygotes likely behave in an uncoordinated fashion and lack proper phototactic movement because the cell reacts to the light stimulus twice per rotation, and therefore cannot properly orient itself toward or away from the light source. Although phototaxis was not verified in the *C. reinhardtii* cells shown in Fig. 6b and c, uncoordinated phototaxis of such cells (Supplementary Fig. 3a) is supported by previous research using an eyeless strain of the disk-shaped biflagellate green alga *Mesostigma viride*[40], which has a rudimentary eyespot that lacks a normal carotenoid pigmented area and does not reflect incident light. The cell of the eyeless strain reacts to light stimuli twice per rotation (once per 180° rotation) due to the lack of a normal eyespot, which reflects and blocks the incident light that penetrates the cell body, and exhibits diaphototaxis (movement perpendicular to the direction of the incident light beam) (Supplementary Fig. 3b), while the cell of a wild-type strain reacts to light stimuli once per rotation due to the presence of an intact eyespot and exhibits positive phototaxis (Supplementary Fig. 3c). Although the eyeless cell is not a planozygote and the direction of phototaxis differs between them, the frequency and interval of the reaction to light stimuli per rotation is similar to that of the planozygotes shown in Fig. 6b and c, suggesting that such planozygotes cannot exhibit proper phototaxis. Consequently, side-by-side alignment of two eyespots on the same side of the cell appears to be a prerequisite for proper phototaxis of the planozygote, while other configurations cannot support proper

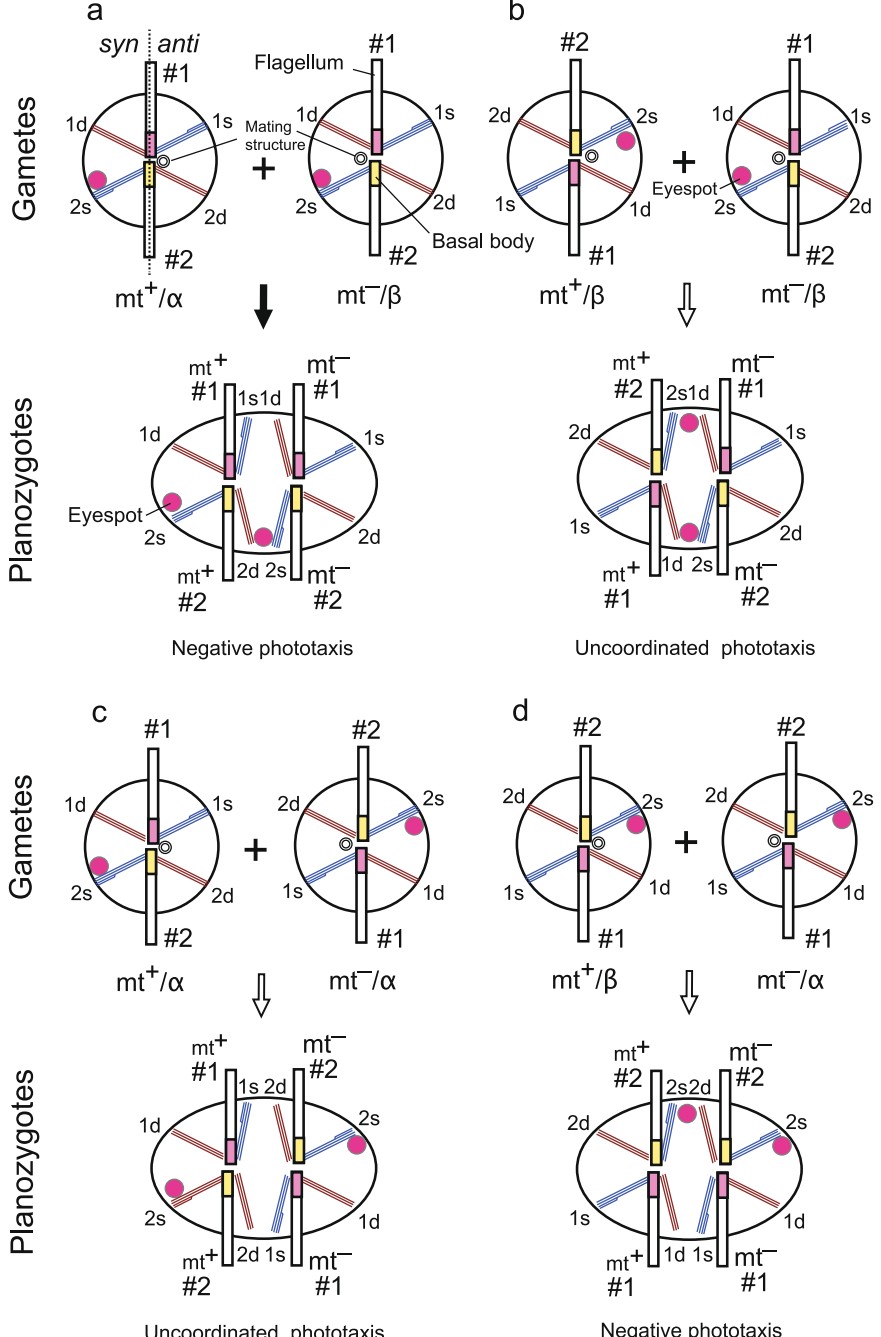

**Fig. 6 Schematic diagrams of possible gamete fusion patterns of *C. reinhardtii*. a** Gamete fusion pattern in *C. reinhardtii*. **b** If the mt⁺ gamete has a mating structure on the same side of the eyespot as the beat plane, the eyespots are distributed on both sides of the cell after fusion with the normal mt⁻ gamete. **c** If the mt⁻ gamete has a mating structure on the side of the beat plane opposite the eyespot, the eyespots are distributed on each side of the beat plane after fusion with the normal mt⁺ gamete. **d** If mt⁺ and mt⁻ gametes have a mating structure on the same or opposite side of the beat plane as the eyespot, respectively, the eyespots are aligned on the same side of the cell after gamete fusion. This is a reverse case of **a**. All cells were viewed from the cell anterior. Predicted outcomes of planozygote phototaxis are shown below the diagrams. *#1*: no. 1 flagellum, *#2*: no. 2 flagellum, *1d*, *1s*, *2d* and *2s*: 1d, 1s, 2d and 2s microtubular roots, respectively, *α*: type α, *β*: type β.

phototaxis, as suggested by Holmes and Dutcher[4]. Third, mt⁺ and mt⁻ gametes have the mating structure on the same side and opposite side of the beat plane as the eyespot, respectively (Fig. 6d). In this case, the planozygote would have parallel flagellar pairs (two no. 1 and two no. 2 flagellar pairs) and two eyespots on the same side of the cell. Such planozygotes would exhibit proper phototaxis, but the gamete fusion pattern shown in Fig. 6d, in which fusion of mt⁺ and mt⁻ gametes occur between

the same side and opposite side of the beat plane as the eyespot, respectively, has not been adopted during the evolution of *C. reinhardtii*. For these reasons, if MSP is not regulated by *MID* and a gamete could fuse with another gamete on the *syn* or *anti* side at random, coordinated alignment of four flagella and two eyespots would not occur in 50% of planozygotes. Consequently, almost half of the planozygotes would not exhibit proper phototaxis and would be unable to identify a suitable place for settlement.

Therefore, tight association between *MID* and asymmetric positioning of the mating structure has probably evolved in *C. reinhardtii* to ensure the proper arrangement (Fig. 6a), and exclude other combinations (Fig. 6b–d), of four flagella and two eyespots in the planozygote. However, why the fusion pattern shown in Fig. 6d has not been selected is unknown.

Although asymmetric positioning of the mating structure (type α and β MSPs) likely evolved in *C. reinhardtii* to ensure the coordinated alignment of flagella and eyespots in the planozygote for phototaxis (Fig. 6a), this possibility has not been empirically verified in *C. reinhardtii*. However, these gamete and planozygote traits are prevalent in many chlorophyte species[41,42], which usually produce biflagellate gametes with one eyespot and quadriflagellate planozygotes with two eyespots, and negative phototaxis is common in the planozygotes of *C. reinhardtii* and ulvophycean species (Supplementary Table 1). These findings suggest that asymmetric positioning of the mating structure, coordinated alignment of planozygote flagella and eyespot(s), and phototaxis are conserved in chlorophytes. Therefore, it is likely that sufficient selective pressure acts on the gamete and planozygote stages to maintain these traits during chlorophyte evolution, although causal relationships among these traits has not been verified. Nevertheless, to improve our understanding of the adaptive benefits of MSP, it is necessary to determine empirically how planozygote motility might influence zygote fitness and how strong the selection pressure might be for phototaxis in chlorophyte planozygotes. However, we cannot exclude the possibility that gamete MSP configuration is important for other cellular functions.

Although *MID* orthologs have been found in other volvocine species, such as *Gonium pectorale*[25], *Pleodorina starrii*[24] and *Volvox carteri*[26], as mt− or male-specific genes, the spatial positioning of the mating structure has not been determined in these species. Therefore, it is unknown whether *MID* orthologs are involved in the positioning of the mating structure in other volvocine species. In contrast, mating type- or sex-specific positioning of the mating structure is established in ulvophycean green seaweeds examined to date, in which male and female gametes always belong to the type α and β gametes[42], respectively, but the mating type- or sex-determining gene has not been identified. Nevertheless, this relationship between the mating type (sex) and MSP (type α and β) was confirmed in the slightly anisogamous species *U. prolifera* and *U. partita*, respectively, using the mt− (male) and mt+ (female) specific genes in the mating type locus, such as *PRA1m* (mt−), and *PRA1p* (mt+)[21,22], which were identified using genome sequencing for both mating types of *U. partita*[21] and confirmed as mating type-specific genes by polymerase chain reaction (PCR)-based genotyping in *U. prolifera*[22]. The presence or absence of a mating type-specific gene is correlated with the MSP (type α or β). Gametophytes that produced type α and β gametes always had mt− and mt+-specific genes, respectively, suggesting that MSP in *Ulva* is probably regulated by a mating type locus and mating type (sex)-determining gene, as in *C. reinhardtii*. Presumably, this is also true for other ulvophycean species because male and female gametes always exhibit the type α and β phenotypes, respectively[42]. However, the relationships between MSP and other mating type (sex)-specific traits (e.g. cytoplasmic inheritance of organelles) is not always invariable among *C. reinhardtii*, *Ulva* and other ulvophycean species; chloroplast DNA is usually inherited by progeny from type α gametes (mt+) in *C. reinhardtii*[43], type β gametes (mt+) in *U. partita*[17,44] and type β gametes (females, mt+) in other ulvophycean species (e.g. *Bryopsis maxima* and *Acetabularia caliculus*)[9,45–48], whereas mitochondrial DNA is inherited by progeny from type β gametes (mt−) in *C. reinhardtii*[49], type α/β gametes (mt+) in *U. partita*[50], and type β gametes (females) in

*B. maxima*[45,46] (Table 2), suggesting that the genetic pathway that determines these mating type-specific traits in *Ulva* and other ulvophycean species is not necessarily the same as that of *C. reinhardtii*. Despite these differences, the dominant mating type was commonly associated with a particular MSP phenotype (type β) in *C. reinhardtii* and *U. prolifera* (Table 2), which was clarified using heterozygous diploid gametes harbouring both mt+ and mt− genomes. These gametes displayed mt+ and mt− respectively in *U. prolifera*[22] and *C. reinhardtii*[32], and exhibited the type β phenotype, indicating that these mating types are dominant in each species and commonly associated with the type β phenotype. Because *C. reinhardtii* and *U. prolifera* are distantly related (Chlorophyceae and Ulvophyceae, respectively), these findings suggest that this association was selected for and conserved during chlorophyte evolution, regardless of the underlying mating type (sex) determination mechanism. However, it is necessary to examine more species to conclude the importance of this phenomenon.

Overall, the findings of this study indicate that gamete MSP dimorphism is regulated by the mating type-determining gene *MID*, in a manner similar to other mating type-specific traits in *C. reinhardtii*. Considering the prevalence of mating type- or sex-specific asymmetric positioning of the mating structure in isogamous and anisogamous chlorophyte species and the tight association between the MSP phenotype (type α and β) and particular mating type-specific genes in *Ulva* and sex in other ulvophycean species, we expect that MSP regulation by mating type (sex)-determining genes will be detected in *Ulva* and other chlorophytes, as in *C. reinhardtii*.

## Methods

**Strains and induction of gametogenesis**. The *C. reinhardtii* strains CC-124 (mt−), CC-125 (mt+), CC-127 (diploid mt−), CC-3712 (mid mt−) and CC-3947 (*nic7*, *thi10*, with a *MID* transgene, mt+ *T-MID*) were obtained from the Chlamydomonas Resource Center, University of Minnesota (Supplementary Table 2). CC-3712 (mid mt−) is the deletion mutant; in this mutant, 8–9 kb of segment 3, 10–12 kb of segment 4, and ~10 kb of intersegment DNA of the R domain of the mt− mating type locus, in which *MID* is located, are deleted, and it behaves as mt+ (pseudo-plus)[27]. Although, in addition to *MID*, other sequences are deleted in CC-3712, the pseudo-plus phenotype of CC-3712 is presumably caused by deletion of *MID* because this mutation was complemented by transformation with *MID*[51]. CC-3947 was originally mt+, but behaves as mt− (pseudo-minus) because a 3.5 kb ApaI fragment from the intersegment DNA, in which *MID* is located, and segment 4 of the mt− mating type locus[23] were introduced into the genetic background of mt+ by *MID* transformation[28]. Both mating type-reversed strains exhibited apparently normal growth under laboratory conditions. To induce gametogenesis, all strains were cultured in Tris acetic acid phosphate (TAP) liquid medium for 3–5 days at 23 °C under continuous light (~50 μmol photons/m²/s), and subsequently on TAP 1/2 N (half strength nitrogen) agar for 4–5 days[52]. Cells were recovered from the agar and incubated in nitrogen-free medium[13] for 4–5 h at 25 °C under continuous illumination.

**Detection of *MID***. To ascertain the presence or absence of *MID* and *FUS1* (encoding the mt+-specific glycoprotein[29]) in the wild-type and mating type-reversed strains, a PCR-based method was applied using mating type-specific primers pairs. Total DNA was extracted from 125 mL of a 4-day-old culture of *C. reinhardtii* using the DNeasy Plant Mini Kit (Qiagen, Germanton, MD, USA). The oligonucleotides MTM3F (5′-CGACGACTTGGCATCGACAGGTGG-3′) and MTM3R (5′-CTCGGCCAGAACCTTTCATAGGGTGG-3′) were used for amplification of *MID*, and MTP2F (5′-GCTGGCATTCCTGTATCCTTGACGC-3′) and MTP2R (5′-GCGGCGTAACATAAAGGAGGGTCG-3′) were used for amplification of *FUS1*[53]. To amplify the sequences, TaKaRa Ex Taq Hot Start Version (TaKaRa Bio Inc., Shiga, Japan) was used following the manufacturer's protocol. The PCR programme comprised 35 cycles of amplification, each consisting of denaturation at 95 °C for 10s followed by annealing/elongation at 68 °C for 1 min. The obtained DNA fragments were resolved by agarose gel electrophoresis and visualised by staining with GelRed (Biotium, Fremont, CA, USA). As expected, *FUS1* was detected in CC-125, CC-127 and CC-3947 but not in CC-124 or CC-3712. *MID* was detected in CC-124, CC-127, and CC-3947 but not in CC-125 or CC-3712 (Supplementary Figs. 1, 2).

**Activation of gametes**. To induce elongation of the fertilisation tube from the mt+ mating structure, gametes were activated by incubation with 10 mM dibutyryl-cAMP and 1 mM 3-isobutyl-1-methylxanthine for 60 min at 25 °C[54].

**Table 2 Relationships among MSP, inheritance patterns of chloroplast and mitochondrial DNA, dominant mating type and sex type in *Chlamydomonas reinhardtii*, *Ulva prolifera*, *U. partita* and other ulvophycean species.**

| Species | Sexual reproduction type | MSP (α/β)[a] | | Inheritance pattern of chloroplast DNA[b] | Inheritance pattern of mitochondrial DNA[b] | Dominant mating type | Mating type- or sex-determining gene | Phenotype (α/β) associated with dominant mating type | Reference |
|---|---|---|---|---|---|---|---|---|---|
| | | mt+/female | mt−/male | | | | | | |
| *C. reinhardtii* | Isogamy | α | β | α | β | mt− | *MID* | β | 23,43,49, this study |
| *U. prolifera* | Slight anisogamy | β | α | ND | ND | mt+ | ND | β | 22 |
| *U. partita* | Slight anisogamy | β | α | β | α/β | ND | ND | ND | 17,21,44,50 |
| other ulvophycean species | Isogamy/ anisogamy | β | α | β | β | ND | ND | ND | 9,45-48 |

ND not determined.
[a]α: type α, β: type β.
[b]Uniparental inheritance from type α or β gamete.

**Light microscopy.** One volume of gamete suspension was mixed with one volume of fixative containing 2% glutaraldehyde in nitrogen-free medium. To visualise planozygotes, cells were fixed for 15, 25 and 30 min after mixing mt+ and mt− gametes and observed under a microscope (BHS-RFC; Olympus Optical Co. Ltd., Tokyo, Japan) equipped with differential interference contrast optics. Photographs were obtained using a digital camera (EOS kiss x7; Canon, Tokyo, Japan).

**Immunofluorescence microscopy.** To visualise fertilisation tubules, one volume of activated gamete suspension was mixed with one volume of fixative containing 4% paraformaldehyde, 50 mM piperazine-1,4-bis(2-ethanesulfonic acid)(PIPES), 2 mM $MgSO_4$, and 5 mM ethylene glycol tetraacetic acid (EGTA; pH 7.0) on coverslips coated with 0.1% poly L-lysine (Sigma-Aldrich Co., St. Louis, MO, USA) for 10 min at room temperature. Next, the coverslips were incubated in methanol for 15 min at −20 °C, air-dried, and incubated with 1% bovine serum albumin in phosphate-buffered saline (PBS) for 1 h at room temperature. The coverslips were incubated with an anti-actin antibody (dilution of 1:100, A2066, Sigma-Aldrich Co.) overnight at 4 °C, rinsed with PBS for 30 min, and then incubated with an Alexa 488 goat anti-rabbit IgG (H + L) antibody (dilution of 1:200, Molecular Probes Inc., Eugene, OR, USA) overnight at 4 °C. The cells were washed three times with PBS for 10 min and mounted in Slow Fade Gold (Molecular Probes, Inc.) containing 0.3 µg/mL 4′,6-diamidino-2-phenylindole (DAPI). For double staining of microtubular roots and the fertilisation tubule, one volume of activated gamete suspension was mixed with one volume of fixative containing 5% paraformaldehyde, 3% Triton X100, 50 mM PIPES, 2 mM $MgSO_4$, and 5 mM EGTA (pH 7.0) on coverslips coated with 0.1% poly-L-lysine for 10 min at room temperature. Next, the coverslips were incubated in methanol for 5 min at −20 °C, air dried, and incubated with 1% bovine serum albumin in PBS for 1 h at room temperature. Next, the coverslips were incubated with a monoclonal anti-acetylated tubulin antibody (dilution of 1:100, 6–11B-1; Sigma-Aldrich Co.) and anti-actin antibody (dilution of 1:00, A2066; Sigma-Aldrich Co.) overnight at 4 °C, rinsed with PBS for 30 min, and then incubated with an Alexa Fluor 555 goat anti-mouse IgG (H + L) and Alexa 488 goat anti-rabbit IgG (H + L) (dilution of 1:200, Molecular Probes Inc.) overnight at 4 °C. The cells were washed three times with PBS for 10 min and mounted in Slow Fade Gold. The immunofluorescence control was prepared by omitting the primary antibody. Observations were made using an epifluorescence microscope (BHS-RFC; Olympus Optical Co. Ltd.) equipped with differential interference contrast optics. Photographs were obtained using a digital camera (EOS kiss x7). The length of microtubular roots was measured using ImageJ software (National Institutes of Health, Bethesda, MD, USA).

**Field emission scanning electron microscopy.** To visualise fertilisation tubules, one volume of activated gamete suspension was mixed with one volume of fixative containing 2% glutaraldehyde in nitrogen-free medium. Mating pairs were fixed 15 min after mixing the gametes. Fixed cells were placed on a Nucleopore polycarbonate membrane (Whatman Japan KK, Tokyo, Japan) coated with 0.1% poly-L-lysine (Sigma Chemical Co.) and incubated at 4 °C overnight. Next, they were washed with 10 mM 4-(2-hydroxyethyl)-1-piperazineethanesulfonic acid (HEPES) buffer (pH 7.0) at room temperature. Post-fixation was performed in 1% $OsO_4$ in 10 mM HEPES buffer overnight at 4 °C. Next, the cells were washed with 10 mM HEPES buffer and treated with 0.1% tannic acid in 10 mM HEPES buffer for 15 min at room temperature, washed with 10 mM HEPES buffer, and treated with 1% $OsO_4$ in 10 mM HEPES buffer overnight at 4 °C. After dehydration through a graded ethanol series, the cells were infiltrated with t-butyl alcohol and lyophilised at 4 °C. Samples were coated with $OsO_4$ in a Neoc osmium coater (Meiwafosis Co., Ltd, Osaka, Japan) or with platinum–palladium using the E1045 Ion Sputterer (Hitachi Corp., Tokyo, Japan) and observed using a field emission scanning electron microscope at 2 kV (S5000; Hitachi Corp.) or 5 kV (JSM6330F; JEOL, Tokyo, Japan). All scanning electron micrographs except those in Fig. 2g–i were obtained using the JSM6330F.

**Photographs.** Photographs were exported to Adobe Photoshop CS6 (Adobe Systems Inc., San Jose, CA, USA), ImageJ software (National Institutes of Health, Bethesda, MD, USA) or Pixelmator Pro 2.1.3 Coral (Pixelmator Team, Vilnius, Lithuania) and mounted using Affinity Designer 1.8.3 (Serif [Europe] Ltd., Nottingham, UK).

**Statistics and reproducibility.** Statistical analyses were performed using StatPlus (AnalystSoft Inc., Walnut, CA, USA) and Mac Statistical Analysis ver. 2.0 (Esumi, Tokyo, Japan) software and are detailed in the figure legends.

**Reporting summary.** Further information on research design is available in the Nature Portfolio Reporting Summary linked to this article.

## Data availability

The authors declare that the source data supporting the findings are provided in the paper, the Supplementary Information and Supplementary Data 1 (including data used to plot figures).

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

## Acknowledgements

This work was supported in part by a KAKENHI Grant-in-Aid for Scientific Research (C) (no. 22570084), (B) (nos. 20370025, 22370024), and Challenging Exploratory Research (no. 15K14579). We thank Ryota Akimoto for performing the fluorescence microscopy of microtubular roots.

## Author contributions

S.M. conceived the study and wrote the manuscript. R.I., S.M. and M.O. performed the experiments. S.M. and T.N. performed the FE-SEM analysis. S.M., K.I., T.Y. and S.K. discussed the results and their implications. All authors participated in the editing of the text and approved the final manuscript.

## Competing interests

The authors declare no competing interests.
