## [Peer Review File · Communications Biology]

Reviewers' comments:

Reviewer #1 (Remarks to the Author):

This is a tight, well-crafted manuscript showing that the location of the mating structure in *Chlamydomonas reinhardtii* is under control of the MID gene product. The experiments and their interpretations are clearly explained and the results are unambiguous. The manuscript does a nice job of presenting the background for the question being addressed, and appropriately places the results in an evolutionary context.

One minor comment is that the sentence on line 103 that begins with "We noted . ." is confusing, and actually doesn't make sense to me. If the authors feel that the sentence indeed is written as they intended, then it should be revised to make it clear to most readers.

Reviewer #2 (Remarks to the Author):

Innami et al. investigated an important feature of sexual gametes, the mating structure positioning (MSP) for its regulation between two definable types, alpha and beta, in *C. reinhardtii* as a model.

Innami et al. study adds MSP to the short-list of mating-type-regulated phenotypes in *C. reinhardtii*. The study design using the available mating-type-reversed strains is straightforward and worked well. The presented data is of excellent quality, examining MSP using FE-SEM and immunostaining. Provided schemes in Fig.1 are beneficial for understanding the three dimensional MSP phenotypes. Presented figures/tables support the conclusion that MSP alpha and beta types are controlled by the same regulatory network governing mating types in isogamous species, *C. reinhardtii*.

However, the manuscript is found difficult-to-read. The following includes my comments.

1. In the Discussion, present the *Ulva* case using published data and descriptive phenotypes, helping readers compare the *Ulva* case with *C. reinhardtii* case.

In particular, 1) provide published evidence for the statement that type beta is the dominant type in *Ulva prolifera*, and 2) explain the *Ulva prolifera*'s zygote phenotype, where MSP is deemed necessary.

2. Suggest avoiding the use of gamete type for MSP type alpha and beta. 'Gamete type' is a rather generic term that can be used more broadly than the author's primary usage, MSP type. An alternative term is MSP, abbreviating mating structure position.

3. The authors present a compelling argument that the zygotes' phototactic motility (or planozygotes) can be the reason for evolving dimorphic MSP types in gametes. Then, the fertilized egg in oogamous species is immotile; therefore, MSP coordination becomes irrelevant. Given that, I suggest revising the authors' arguments regarding the timing of MSP regulation and the potential regulation of MSP in oogamy species.

4. Tables 2 and 3 may be combined for clarity.

5. Figure 6 may be better only with alpha and beta labels. Recommend to add the predicted outcome (positive/negative/uncoordinated phototaxis) and the example species that possess each type.

P7, in Discussion. CC-125 is a 137c strain, expecting nearly identical genetic makeup.

Reviewers #3-4 (Remarks to the Author):

Innami et al.

This is a very well written and presented manuscript and I commend the authors on the clarity with which they have presented the text and figures.

The evolution of anisogamy from isogamous ancestral forms remains an intriguing area of evolutionary study. The focus of these studies has narrowed down to the green algae where both isogamous and anisogamous forms exist. In the isogamous forms, the presence of mating types and associated gamete dimorphism suggest that these traits preceded the evolution of the sexes. In *Chlamydomonas*, the mating types (mt+ and mt-, where mt+ grows the mating tubule to connect to the mating structure of the mt- gamete) are associated with dimorphism in the position of the cell fusion site with respect to the arrangement of the flagella and eyespot.

Although the mating type locus MID is well studied, the effect of MID on the position of the mating site (that either grows or receives the mating tubule) was hitherto unknown.

Here, Inammi et al used stains of *Chlamydomonas* that are mating type reversed through the deletion of MID or introduction of MID as a novel way of investigating the role of MID on the position of the mating structure.

Their experimental protocols are clear and statistical analyses are sound and statistically conclusive.

The presence of MID is clearly associated with the location of the mating structure, demonstrating that the growth (or not) of the mating tubule and the location of the mating structures, are both controlled by MID.

Major concerns

There are several areas where the significance of this study has not been demonstrated or are unclear.

First, although gamete type (the position of the mating structure on the cell; alpha or beta) is dimorphic the significance of this from the point of view of the evolution of anisogamy seems less clearly articulated. The importance of mating type is clearly important because mating types must mate with those of the opposite sign, this establishes the arena for gamete competition and the evolution of sex roles. This is not appear to be true however for the gamete type (alpha or beta), since arguments are constructed about selection on the planozygote to generate disruptive selection at this stage. Thus it is not clear how important gamete type is for the evolution of anisogamy.

Second, from an evolutionary perspective the arguments relating to the selection on planozygote phase of the life-cycle are, without further evidence, rather weak. The argument that the phototaxis of the planozygote will be compromised is persuasive, however to maintain strong linkage between traits selection must be persistent and strong. To our knowledge, the planozygote exists for only a short period of time – is selection on this phase strong enough to maintain the observed pattern of linkage between the position and the growth of the mating tubule? This section seems rather speculative without evidence for the strength of selection.

Finally, the conclusion on the relation between the sex-determining gene, gamete types (alpha and beta) and the evolution of anisogamy in the chlorophytes is confusing. The authors state that the gamete type (alpha and beta), has a preference for a specific sex type and that the system determining the gamete type (alpha or beta) were selected and conserved during chlorophyte evolution, and therefore emerged before males and females. This seems to contrast with the results that the position of the mating structure and consequently, the gamete type in *Chlamydomonas* are determined by the sex-determining gene MID but that there is a reversed relationship in *Ulva* between mating type and gamete type.

Minor points

L32. such as *Ulva prolifera*

L44. behaviour 3, and so these dimorphisms must have preceded....

L72 – unclear why this sentence begins with 'nevertheless'.

L103: "We noted that the fertilisation tubule elongated from the mt+ mating structure instead of the mating structure itself." The sentence is not very clear.

L187: Strain 137c is the same as CC-125 (<https://www.chlamycollection.org/product/cc-125-wild-type-mt-137c/>)

L228 here there is a profusion of descriptors of different facets of the cells biology or structure that might be assisted by including explanations in parentheses or being more consistent - sex (aka mating type) and gamete type (alpha or beta).

L230: Reference 40 is missing from the references list.

Dear Reviewers

Thank you for your valuable comments and suggestions. We have studied the comments carefully, made the necessary corrections and rewrote the manuscript according to the reviewer's suggestions, which are highlighted within the document by red color text (red color text for the changes according to the reviewer's comment and blue for the rewrite parts). The followings (*italics*) are our response to reviewer's comment.

Reviewer #1 (Remarks to the Author):

(1) One minor comment is that the sentence on line 103 that begins with "We noted.." is confusing, and actually doesn't make sense to me. If the authors feel that the sentence indeed is written as they intended, then it should be revised to make it clear to most readers.

*Thank you for your suggestion. We deleted this sentence and rewrote this part as follows "However, CC-3712 (mid mt^-) gametes **could produce fertilisation tubules but did not fuse with mt^- or mt^+ gametes because this strain lacks the cell-adhesion gene $FUS1^{29}$ (Supplementary Fig. 1). **Therefore, we directly observed the fertilisation tubule elongation from the mt^+ mating structure.**" (page4, lines107-110). (red color text for the changes according to the reviewer's comment).***

Reviewer #2 (Remarks to the Author):

1. In the Discussion, present the Ulva case using published data and descriptive phenotypes, helping readers compare the Ulva case with *C. reinhardtii* case.

In particular, 1) provide published evidence for the statement that type beta is the dominant type in *Ulva prolifera*, and 2) explain the *Ulva prolifera*'s zygote phenotype, where MSP is deemed necessary.

*Thank you for your suggestions. We provided published evidence for (1) the statement that type beta is the dominant type in *Ulva prolifera* (page11, lines327-330, highlighted within the document by red color text) and explained the zygote phenotype of *Ulva* including *U. prolifera* in the Discussion as follows, "For example, the planozygote of ulvophyceae...." (page11, lines302-307).*

2. Suggest avoiding the use of gamete type for MSP type alpha and beta. 'Gamete type' is a rather generic term that can be used more broadly than the author's primary usage, MSP type. An alternative term is MSP, abbreviating mating structure position.

Thank you for your suggestion. We adopted 'MSP' for "mating structure position" and defined the gamete that has MSP (α and β) as type α and β gamete in the "Introduction", respectively, as follow, "In this study, we used the term...." (page3, lines58-60). We used MSP and gamete type (α and β) appropriately in the manuscript.

3. The authors present a compelling argument that the zygotes' phototactic motility (or planozygotes) can be the reason for evolving dimorphic MSP types in gametes. Then, the fertilized egg in oogamous species is immotile; therefore, MSP coordination becomes irrelevant. Given that, I suggest revising the authors' arguments regarding the timing of MSP regulation and the potential regulation of MSP in oogamy species.

Thank you for your suggestion. According to your suggestion, we revised our arguments regarding the timing of MSP regulation and the potential regulation of MSP in "Discussion" (page13, lines371-382). We also added new Figure (Fig. 9 in revised manuscript) to show the timing of MSP regulation during chlorophyte evolution.

4. Tables 2 and 3 may be combined for clarity.

Thank you for your suggestion. According to your suggestion, we combined Table 2 and 3 and made new Table 2.

5. Figure 6 may be better only with alpha and beta labels. Recommend to add the predicted outcome (positive/negative/uncoordinated phototaxis) and the example species that possess each type.

Thank you for your suggestion. According to your suggestion, we deleted MID from Figure 6 (Fig. 8 in revised manuscript) and added the predicted outcome (positive/negative/uncoordinated phototaxis) and the example species (page31, lines812-813).

P7, in Discussion. CC-125 is a 137c strain, expecting nearly identical genetic makeup.

Thank you. We have changed the description as follows. " One possibility is that the position of the mt^+ mating structure varies among strains or laboratory cultures derived from the same strain, as this and the previous studies used the same wild-type strains (CC-125 (137c) in this study, 137c in Goodenough and Weiss¹⁰, and C-541 (137c) in Miyamura et al.⁸), while other research involved a different strain (RC3⁴⁰)."(page9, lines 243-246)

Reviewers #3-4 (Remarks to the Author):

Major concerns

There are several areas where the significance of this study has not been demonstrated or are unclear.

First, although gamete type (the position of the mating structure on the cell; alpha or beta) is dimorphic the significance of this from the point of view of the evolution of anisogamy seems less clearly articulated. The importance of mating type is clearly important because mating types must mate with those of the opposite sign, this establishes the arena for gamete competition and the

evolution of sex roles. This is not appear to be true however for the gamete type (alpha or beta), since arguments are constructed about selection on the planozygote to generate disruptive selection at this stage. Thus it is not clear how important gamete type is for the evolution of anisogamy.

Thank you for your valuable comments. We have revised the manuscript and described our argument about the significance of the “mating structure position” (MSP in the text) (α and β) from the point of view of the evolution of anisogamy from isogamy in the chlorophyte algae in “Discussion” (page12, line338-page13, line391).

Second, from an evolutionary perspective the arguments relating to the selection on planozygote phase of the life-cycle are, without further evidence, rather weak. The argument that the phototaxis of the planozygote will be compromised is persuasive, however to maintain strong linkage between traits selection must be persistent and strong. To our knowledge, the planozygote exists for only a short period of time – is selection on this phase strong enough to maintain the observed pattern of linkage between the position and the growth of the mating tubule? This section seems rather speculative without evidence for the strength of selection.

*Thank you for your valuable comments We performed additional experiments to address whether there is sufficient selective pressure on the planozygote stage to maintain the sex-specific arrangement of the mating structure. For this purpose, we noticed the significance of the asymmetric arrangement of the mating structure for the proper arrangement of eyespots in the planozygote of *C. reinhardtii*. Ideally, we should make the gamete that has disordered arrangement of the mating structure and evaluate the effect of the disorder for the swimming and phototactic behavior of the planozygote. However, it was impossible at present. So instead, we made the planozygotes with disordered arrangement of the eyespots using wild types (CC-124, CC-125 and CC-621) and mutant strains (CC-4304, CC-4320 and CC-4455) with multiple eyespots, and observed the phenotype and phototactic behavior of the planozygotes (Figs. 6, 7, Supplementary Figures 3, 4, 5 and Table S1, S2). Results of the experiments, which showed that alignment of two eyespots on the same side of the planozygote cell is necessary for negative phototaxis, and discussion follow from the results of the experiments were described in the “Results” (page6, line167-page8, line218) and “Discussion” section (page10, line270-page11, line309).*

Finally, the conclusion on the relation between the sex-determining gene, gamete types (alpha and beta) and the evolution of anisogamy in the chlorophytes is confusing. The authors state that the gamete type (alpha and beta), has a preference for a specific sex type and that the system determining the gamete type (alpha or beta) were selected and conserved during chlorophyte evolution, and therefore emerged before males and females. This seems to contrast with the results that the position of the mating structure and consequently, the gamete type in *Chlamydomonas* are

determined by the sex-determining gene MID but that there is a reversed relationship in *Ulva* between mating type and gamete type.

Thank you for your valuable comments. As you pointed out, the description on the relation between the sex-determining gene, gamete types (alpha and beta) and the evolution of anisogamy in the chlorophytes is confusing. Thus, we rewrote the manuscript to clarify our argument for this relationship in chlorophytes (page12, lines338-344).

Minor points

L32. such as *Ulva prolifera*

L44. behaviour 3, and so these dimorphisms must have preceded....

Thank you for your suggestion. We rewrote the manuscript according to your suggestions (L32, page2, line32; L44, page2, line44).

L72 – unclear why this sentence begins with ‘nevertheless’.

According to your comment, we rewrote the manuscript as follows. “To elucidate this point, ulvophycean algae provide insights into the genetic background of this trait....” (page3, line76).

L103: “We noted that the fertilisation tubule elongated from the mt+ mating structure instead of the mating structure itself.” The sentence is not very clear.

*We changed the description as follows to clarify the sentence. “However, CC-3712 (mid mt⁻) gametes **could produce fertilisation tubules** but did not fuse with mt⁻ or mt⁺ gametes because this strain lacks the cell-adhesion gene *FUSI*²⁹ (Supplementary Fig. 1). **Therefore, we directly observed the fertilisation tubule elongation from the mt⁺ mating structure.**” (page4, lines107-110)*

L187: Strain 137c is the same as CC-125

<https://www.chlamycollection.org/product/cc-125-wild-type-mt-137c/>

Thank you. According to your suggestion, we have changed the sentence as follows. “One possibility is that the position of the mt⁺ mating structure varies among strains or laboratory cultures derived from the same strain, as this and the previous studies used the same wild-type strains (CC-125 (137c) in this study, 137c in Goodenough and Weiss¹⁰, and C-541 (137c) in Miyamura et al.⁸), while other research involved a different strain (RC3⁴⁰).” (page9, lines243-246)

L228 here there is a profusion of descriptors of different facets of the cells biology or structure that might be assisted by including explanations in parentheses or being more consistent - sex (aka mating type) and gamete type (alpha or beta).

Thank you. We defined the “mating structure position” (MSP) and the gamete type (type α and β) in the “Introduction” (page3, lines58-60) to clarify the meaning of these terms (gamete type and

*mating structure position(MSP))in this manuscript according to the suggestion of Reviewer#2 and changed the sentence as follows. “Nevertheless, this relationship between **the sex (mating type) and MSP (type α and β)** was confirmed....”(page11, line316)*

L230: Reference 40 is missing from the references list.

Sorry. We revised the reference list.

Reviewers' comments:

Reviewer #2 (Remarks to the Author):

The revised manuscript by Innami et al. fully addressed the reviewer's comments. To improve clarity, I recommend the following to be revised before publication.

L76. "this point" needs to be specified for clarity.

L98-100. "our finding" may be elaborated further, so that readers can understand why the findings help to explain the evolution of two sexes.

- Besides, based on the manuscript's findings, I do not see how the manuscript improves our understanding of the evolution of anisogamy.

L103. CC-3947 (mt(+)-). The genotype description needs to be amended. (mt+ T-MID) may be acceptable.

L180-185/L194-198/L200-202. These three sentences contain too many details in the text. They would be much better legible in a table form. I recommend to make them brief in the text, and they are referred to a table that contains crosses, phenotypes, and the numbers of observations. The table may be a supplemental table or included in Figure 6.

L283. Suggest to describe "the gamete fusion pattern" in words in addition to referring to Figure 8d, which will greatly help readers.

Reviewer #5 (Remarks to the Author):

I read the manuscript paying attention to the points raised by Reviewer 3.

The data in this paper clearly support the conclusion that the mating type determining gene MID in *Chlamydomonas* governs formation of the two types of mating structures (MSPs) termed α and β . This is a new finding. However, like Reviewer 3 from the previous round, I had difficulty following the evolutionary arguments about how MSPs relate to the emergence of either two mating types, and the eventual emergence of two sexes. It seems like the authors are generally overstating the importance of MSP dimorphism in the evolution of sex, and most of their evolutionary arguments remain unclear to me as they were to Reviewer 3.

Regarding the points raised by Reviewer 3

"First, although gamete type . . ." This point was about the roles proposed for MSPs in the evolution of anisogamy. I could not follow the logic in the revised manuscript. My best guess is that the authors were saying that MSPs were responsible for emergence of two mating types (or sex roles) which then led to the preconditions for anisogamy/oogamy. However, I did not find this argument compelling, especially since MSPs appeared to have evolved in *Chlamydomonas* and *Ulva* through autosomal genes that came under the control of the mating type gene MID secondarily. The authors only compound the confusion by suggesting that MSP type is a trait which became genetically linked with mating types, which is the opposite to what they found. If the α and β MSPs were genetically linked to MT, then the sex reversal experiments would have given a different result. I think part of the problem is not just the unclear usage of linkage, but an unclear story about how sexual reproduction and mating types evolved as the authors envision. Did it start with a population of cells that could all fuse with each other? What are the proposed genetics of α and β MSPs and how did they evolve without a genetic switch to toggle the two types? If there was a genetic switch, then one could say that mating types were a precondition to α and β MSPs, and not the reverse. Figure 9 tries to illustrate the author's arguments/speculation, but I could not follow it.

"Second, from an evolutionary perspective . . ." The point raised is that the evidence for strong selection on planozygote phototaxis seems dubious. Not sure I agree. Clearly the cells go to the trouble of ensuring that they fuse so that eyespots can be aligned, and this trait has been

maintained in distant algal species. In any case, we don't know how strong the selection might be for phototaxis in planozygotes since we don't know how this motility might influence fitness of the zygote (perhaps chances of germination or completion of development depend on light?). I also think it is fair to say that phototaxis is not even close to an essential trait for *Chlamydomonas* zygotes, at least not in the laboratory, so it is not hard to envision MSP dimorphism evolving secondarily after mating types. Mating types do control some far more important traits such as agglutination, zygote differentiation factors, and production of complementary membrane fusogens, all of which are essential for the sexual cycle.

"Finally, the conclusion on the relation . . ." The point raised is that the description of the relationship between MSP type and evolution of anisogamy in the Chlorophytes is confusing (similar to the first point). I still find it confusing. The authors repeatedly use the term dominant MSP, but in fact there is no genetic dominance shown for MSPs, but for mating types, which control MSP type. The reversal of the relationship between the dominant mating type and the MSP in ulvophyceans versus volvocine algae has unclear significance. The authors try to tie it to anisogamy, but I could not follow the logic.

Dear Reviewers

Thank you for your valuable comments and suggestions. We have studied the comments carefully, made the necessary corrections and rewrote the manuscript according to the reviewer's suggestions, which are highlighted within the document by red color text (red color text for the changes according to the reviewer's comment and blue for the rewrite parts). The followings (*italics*) are our response to reviewer's comment.

Reviewer #2 (Remarks to the Author):

L76. "this point" needs to be specified for clarity.

Thank you. We clarified "this point" (the genetic background of the asymmetric positioning of the mating structure (type α and β MSPs)) as follows, "Ulvophyceean algae provide insights into the genetic background of the asymmetric positioning of the mating structure (type α and β MSPs)." (L64-65 in revised manuscript)

L98-100. "our finding" may be elaborated further, so that readers can understand why the findings help to explain the evolution of two sexes.

Thank you. We explained "our finding" in the text as follows, "...and showed that MSP acts as a restriction mechanism or constraint preventing mating between related gametes and ensuring the proper gamete pair and planozygote formation for phototaxis. Finally, we compared our findings on the role of MSP with those of previous reports on ulvophyceean species and proposed that MSP (α and β) were responsible for the pre-existence of two mating types in isogamy, which led to the preconditions for anisogamy and oogamy in chlorophytes". (L87-91 in revised manuscript)

- Besides, based on the manuscript's findings, I do not see how the manuscript improves our understanding of the evolution of anisogamy.

To explain how the findings in this study improve our understanding of the evolution of anisogamy, we added the following description at the end of the Introduction, "Finally, we compared our findings on the role of MSP with those of previous reports on ulvophyceean species and proposed that MSP (α and β) were responsible for the pre-existence of two mating types in isogamy, which led to the preconditions for anisogamy and oogamy in chlorophytes". (L88-91 in revised manuscript)

Moreover, to explain how the findings in this study improve our understanding of the evolution of anisogamy, we added the description (red color text) to the “Discussion” that explain the possible role of MSP as a morphological constraint to restrict the number of mating types to two prior to the evolution of anisogamy in chlorophytes. (L374-405 in revised manuscript).

Furthermore, we added the description about the possible relationship between MSP and sex-determining gene in ancestral chlorophytes to the text, which was pointed out by reviewer 5 (L406-424 in revised manuscript).

L103. CC-3947 (mt(+)-). The genotype description needs to be amended. (mt+ T-MID) may be acceptable.

Thank you. We amended the genotype description according to your suggestion (mt+ T-MID). (L95 in revised manuscript)

L180-185/L194-198/L200-202. These three sentences contain too many details in the text. They would be much better legible in a table form. I recommend to make them brief in the text, and they are referred to a table that contains crosses, phenotypes, and the numbers of observations. The table may be a supplemental table or included in Figure 6.

Thank you. We made the descriptions of the crosses, phenotype and numbers of observations briefly in the text and referred to tables (TableS1 and S2). (L173-177/187/189 in revised manuscript)

L283. Suggest to describe "the gamete fusion pattern" in words in addition to referring to Figure 8d, which will greatly help readers.

Thank you. We added the description of the “gamete fusion pattern” in the text. (L286-289 in revised manuscript)

REVIEWER #4:

One important point that I believe should be addressed in the manuscript is why it was not possible to use the planozygote type B on the phototaxis assay. The absence of a planozygote type B on the phototaxis assay is unfortunate as it would be the planozygote type if both mating types had the same MSP.

Thank you. According to your comments, we added the following explanation to the

text, “we could not evaluate the phototaxis of the cells shown in Fig. 8b and c, because the frequency of the planozygote type B was low in all crosses. (L263-264 in revised manuscript)

Another critical point is that the authors state that “These data suggest that the localisation of two or more eyespots on the same side of the cell is necessary for the negative phototaxis of the planozygote” (L216). However, in the crosses CC-4304xCC-4455 and CC-4320xCC-4455, where the planozygotes type D (eyesspots distributed in all segments) were predominant, negative phototaxis was not evident.

Thank you. As you pointed out, our original explanation for the relationship between the eyespot distribution and phototaxis of the planozygotse seems to be misleading. Therefore, we have improved the description and added the additional explanation to the text. (L274-282 in revised manuscript)

Minor comments

L167- 218. I would suggest breaking this part of the text into smaller paragraphs.

Thank you. According to your suggestion, we have divided this part to smaller paragraphs. (L160-207 in revised manuscript)

L180-185. The result from the crosses between strains might be better presented as a table in the manuscript (Table S1) instead of describing the results.

Thank you. We summarized the results in Table S1. (L173-177 in revised manuscript)

L191. “Two eyespots were...” this sentence is not very clear.

Thank you. According to your suggestion, we have tried to clarify the sentence as follows, “In these crosses, the planozygotes....”. (L183-184 in revised manuscript)

L194-198; 200-202. “In CC-4304 x CC-621....” The text is redundant with figure 6n.

Thank you. According to your comment, we made the descriptions of the crosses and phenotype brief in the text and referred to a table (Table S2). (L184-189 in revised manuscript)

L273-276. The argument that the planozygotes shown in Fig. 8b and c would react to light the same as the eyeless strain of *Mesostigma viride* is relatively weak.

Thank you. According to your comments, we added new Figure (Supplementary Fig. 6)

and improved the explanation for the planozygotes shown in Fig. 8b and c deduced from the previous experiments using eyeless strain of Mesostigma viride. (L265-274 in revised manuscript)

L363. “C. reimhardtii”

Sorry. We have corrected the spelling. (L397 in revised manuscript)

L381. “th fertilised egg”

Sorry. We have corrected the spelling. (L435 in revised manuscript)

L382. “for the immotile zygote (egg)” is redundant in the sentence.

Thank you. We have deleted “for the immotile zygote (egg)” from the text. (L436 in revised manuscript)

L794. (Fig.7 legend). “CC621”

Sorry. We have corrected the spelling (CC-621). (L881 in revised manuscript)

Reviewer #5 (Remarks to the Author):

“First, although gamete type . . .” This point was about the roles proposed for MSPs in the evolution of anisogamy. I could not follow the logic in the revised manuscript. My best guess is that the authors were saying that MSPs were responsible for emergence of two mating types (or sex roles) which then led to the preconditions for anisogamy/oogamy. However, I did not find this argument compelling, especially since MSPs appeared to have evolved in Chlamydomonas and Ulva through autosomal genes that came under the control of the mating type gene MID secondarily. The authors only compound the confusion by suggesting that MSP type is a trait which became genetically linked with mating types, which is the opposite to what they found. If the α and β MSPs were genetically linked to MT, then the sex reversal experiments would have given a different result. I think part of the problem is not just the unclear usage of linkage, but an unclear story about how sexual reproduction and mating types evolved as the authors envision. Did it start with a population of cells that could all fuse with each other? What are the proposed genetics of α and β MSPs and how did they evolve without a genetic switch to toggle the two types? If there was a genetic switch, then one

could say that mating types were a precondition to α and β MSPs, and not the reverse. Figure 9 tries to illustrate the author's arguments/speculation, but I could not follow it.

Thank you for your valuable comments.

(1) *As you pointed out usage of "linkage" was unclear in the text, so we replaced "link and linkage" with other terms, e.g., "association" etc., which are highlighted within the document by purple color text.*

(2) I think part of the problem is not just the unclear usage of linkage, but an unclear story about how sexual reproduction and mating types evolved as the authors envision.

We tried to clarify author's argument on the story about how sexual reproduction, mating types and MSP evolved in chlorophytes. For this purpose, we explained in the revised manuscript that MSP probably evolved secondarily after establishment two mating types in the ancestor of chlorophyte and came under the control of sex-determining gene, because this trait is only found in the chlorophytes in spite of the prevalence of two mating types in the chlorophytes, streptophytes and other eukaryotes. We also discussed the other possibility that if ancestor of the chlorophyte had more than two mating types, asymmetry of MSP acted as a morphological constraint to restrict the number of mating types to two.

Furthermore, we also mentioned the other possibilities that if MSP or similar traits are found in other eukaryotes, it is possible that MSP act as a restriction mechanism or morphological constrain underlying the pre-existence of two mating types in isogamy at the early stage of the evolution of mating types in eukaryote. (L406-424 in revised manuscript)

(3) Did it start with a population of cells that could all fuse with each other?

We do not think that MSP starts with a population of cells that could all fuse with each other, but we postulated that MSP starts with a population of cells that two or more mating types as mentioned in the revised manuscript. (L406-424 in revised manuscript)

(4) What are the proposed genetics of α and β MSPs and how did they evolve without a genetic switch to toggle the two types?

We think that MSP did not evolve without a genetic switch but came under the control of particular mating type or sex-determining gene after establishment of two or more mating types as described in the text. (L406-424 in revised manuscript)

(5) We have improved Fig. 9 to clarify that the following statement is reasonable, “MSP likely appeared in the isogamous species that has two mating types in the early evolution of chlorophytes...” because of the reason described above (2) and in the text. (L906-916 in revised manuscript)

“Second, from an evolutionary perspective . . .” The point raised is that the evidence for strong selection on planozygote phototaxis seems dubious. Not sure I agree. Clearly the cells go to the trouble of ensuring that they fuse so that eyespots can be aligned, and this trait has been maintained in distant algal species. In any case, we don’t know how strong the selection might be for phototaxis in planozygotes since we don’t know how this motility might influence fitness of the zygote (perhaps chances of germination or completion of development depend on light?). I also think it is fair to say that phototaxis is not even close to an essential trait for *Chlamydomonas* zygotes, at least not in the laboratory, so it is not hard to envision MSP dimorphism evolving secondarily after mating types. Mating types do control some far more important traits such as agglutination, zygote differentiation factors, and production of complementary membrane fusogens, all of which are essential for the sexual cycle.

Thank you for your valuable suggestions.

*As you pointed out, phototaxis is not even close to an essential trait for *Chlamydomonas* zygotes at least in the laboratory. However, we thought that phototaxis of the planozygotes is probably crucial for the survival of the planozygote under natural condition. To support our argument, we rewrote the manuscript as follows*

(6) As you pointed out, chance of germination of cysts depend on light illumination, so that we rewrote the explanation for the adaptive benefit of the phototaxis of planozygote as follows, “Taking into consideration that cyst development proceeds both in the presence and absence of light while cyst germination is sufficiently induced by.... “. (L301-304 in revised manuscript)

(7) We mentioned another possibility that movement of the planozygote to dimly lit area of the bottom of the pond or in the soil using negative phototaxis and subsequent settlement is an overwintering strategy. (L304-308 in revised manuscript)

- (8) *We added Table S3 that summarized the MSP, number of flagella and eyespot(s)/cell and phototaxis of gametes and planozygotes in the chlorophytes, which show that MSP and negative phototaxis of the planozygote is commonly found in the major orders of the Ulvophyceae, suggesting that these traits have been selected and conserved during the evolution. (L313-322 in revised manuscript)*
- (9) *As you pointed out, we don't know how strong the selection might be for phototaxis in planozygotes and how this motility might influence fitness of the zygote. So, we have mentioned these points in the "Discussion". (L324-327 in revised manuscript)*

"Finally, the conclusion on the relation . . ." The point raised is that the description of the relationship between MSP type and evolution of anisogamy in the Chlorophytes is confusing (similar to the first point). I still find it confusing. The authors repeatedly use the term dominant MSP, but in fact there is no genetic dominance shown for MSPs, but for mating types, which control MSP type.

- (10) *Thank you. As you pointed out, "dominant MSP" is ambiguous, so that we have changed "dominant MSP" to "dominant phenotype of MSP" in the text, because diploid strain containing both mt+ and mt- genome exhibited type β MSP phenotype, indicating that type β is a dominant phenotype.*
- (11) *The reversal of the relationship between the dominant mating type and the MSP in ulvophyceans versus volvocine algae has unclear significance. The authors try to tie it to anisogamy, but I could not follow the logic.*
Thank you. We improved the sentence describing the reversal of the relationship between the MSP and anisogamy in ulvophyceans versus volvocine algae. We also compared the reversal of the relationship between the MSP and mating type (sex) with the reversal of sex specific structural trait or sex roles in cave insect and mentioned the significance of the reversal of sex-specific trait in isogamous and anisogamous organisms. (L358-373)
- (12) *Furthermore, as you pointed out, the reason and significance of the reversal of the relationship between the dominant mating type and the MSP in ulvophyceans versus volvocine algae is still unclear. Therefore, we mentioned these points at the end of the paragraph as follows, "It is also unclear why the relationship between the mating type and phenotype of MSP is reversed between C. reinhardtii and U. prolifera irrespective of whether they are dominant (Table 2)". (L355-357 in revised manuscript)*
- (13) *The authors try to tie it to anisogamy, but I could not follow the logic.*

Thank you. We tried to clarify the logic (L374-405 red color text in revised manuscript) and improved Fig. 9 as described above (5).

Reviewers' comments:

Reviewer #5 (Remarks to the Author):

This revised manuscript tried to address some of the points of prior reviewers, but the authors seem to remain fixated on discussing topics to which their data don't shed light, particularly the evolution of sexes and the evolutionary relationship of MSP to mating type differentiation. The data in this paper clearly support the conclusion that the mating type determining gene MID in *Chlamydomonas* governs formation of the two types of mating structures (MSPs) termed α and β . Beyond that, the evolutionary arguments are convoluted and not persuasive. If the authors want to make more extended and synthetic arguments and speculations it would be better to do so in a separate review or opinion piece rather than as an add-on to this manuscript.

Major points.

1. I remain puzzled why the authors emphasize and discuss at length the evolution of anisogamy and oogamy starting on line 358 because their data don't really shed light on these types of dimorphism. The lengthy discussion of this topic does not add anything to the manuscript in my opinion. The evolutionary scenario described in Fig. 9 starting on line 437 does not really add anything to understanding the evolution of sexes. MSP dimorphism is one of many molecular traits that differ between mating types and which would have preceded the emergence of anisogamy and oogamy. If asymmetric MSPs never evolved at all I don't see how it would impact our understanding of anisogamy and oogamy.
2. The authors persist in using language that describes beta MSP as dominant eg. line 18 "type β is the dominant phenotype," and lines 84/85, 148, 159. As was pointed out previously, there is no dominance relationship between MSPs. MSPs are under the control of MID or mating type determining genes, and it is these mating type determining genes that display a dominance relationship, not the MSPs.
3. The authors make a point about evolution of MSPs and their relationship to the plus and minus mating types in *Chlamydomonas* versus *Ulva* (lines 355-357). Unfortunately, this is an empty argument as mating type assignment for isogamous algae was previously decided based on somewhat arbitrary criteria. In the case of *Chlamydomonas* it was uniparental chloroplast inheritance that was designated as the maternal parent (mt+), but if mitochondrial inheritance had been used as the criterion then the assignment of mating types would have been reversed. It is interesting that in both species the dominant mating type is associated with the beta-type MSP, but with a sample size of two species there is not much to conclude about this observation.
4. The additional experiments that are meant to test the impact of eyespot placement in planozygotes are flawed and not convincing. The underlying defects of mlt (multi eyespot) mutants are not well understood. While mlt strains have extra eyespots, they may also have cytoskeletal defects or other defects that impact phototaxis independently of the extra eye spots.
5. I don't think that demonstrating phototaxis defects in planozygotes with mismatched eye spots (i.e., those that fuse with eyespots on opposite sides) is necessary. The ability to form asymmetrically placed mating structures in gametes of opposite mating type is conserved, and to me that is a strong argument for MSP dimorphism being under selection. Accordingly, the long speculative paragraph starting on line 297 could be significantly shortened. The problem with testing phototaxis is that even if it could be done convincingly, it is possible that the asymmetric MSP configuration is important for something else, such as proper organization of the meiotic cytoskeleton, so proving/disproving its utility for phototaxis would not rule out other roles where there might be even stronger selection.

Minor Points

6. Please use "sexes" and "mating types" appropriately. *Chlamydomonas* and other isogamous algae do not have two sexes, they have two mating types.
7. Lines 13/14. "based on the sex-specific asymmetric positioning of the mating structure".

Asymmetric position with respect to what?

8. Line 24. Eukaryotes do not generally have two sexes or mating types. While two mating types are most common, there are many examples of multiple mating types. Multiple sexes are not observed. Even with hermaphrodites there are still only two sex roles.

9. Lines 82-84. This is an awkward sentence that needs a rewrite, perhaps splitting into two sentences. Sex-reversed is inaccurate language. The strains are mating-type reversed.

10. Lines 333-6. Unclear sentence. What were the mating specific genes used for? PCR genotyping?

Dear Reviewers

Thank you for your valuable comments and suggestions. We have limited the scope of the manuscript to focus on algal biology and role of *MID* in MSP formation and deleted the discussion about the sex evolution from the manuscript according to your suggestion. We have also studied the comments of reviewer#5 carefully, made the necessary corrections and rewrote the manuscript according to the reviewer's suggestions, which are highlighted within the document by red color text (red color text for the changes according to the reviewer's comment and blue for the rewrite parts). The followings (*italics*) are our response to reviewer's comment.

Reviewer #5 (Remarks to the Author):

This revised manuscript tried to address some of the points of prior reviewers, but the authors seem to remain fixated on discussing topics to which their data don't shed light, particularly the evolution of sexes and the evolutionary relationship of MSP to mating type differentiation. The data in this paper clearly support the conclusion that the mating type determining gene *MID* in *Chlamydomonas* governs formation of the two types of mating structures (MSPs) termed α and β . Beyond that, the evolutionary arguments are convoluted and not persuasive. If the authors want to make more extended and synthetic arguments and speculations it would be better to do so in a separate review or opinion piece rather than as a add-on to this manuscript.

Thank you for your valuable comments. We have deleted the discussion about the sex evolution from the manuscript according to your and editor's suggestion. Concerning evolutionary argument, we would like to make a separate review or opinion pieces according to your suggestion.

Major points.

1. I remain puzzled why the authors emphasize and discuss at length the evolution of anisogamy and oogamy starting on line 358 because their data don't really shed light on these types of dimorphism. The lengthy discussion of this topic does not add anything to the manuscript in my opinion. The evolutionary scenario described in Fig. 9 starting on line 437 does not really add anything to understanding the evolution of sexes. MSP

dimorphism is one of many molecular traits that differ between mating types and which would have preceded the emergence of anisogamy and oogamy. If asymmetric MSPs never evolved at all I don't see how it would impact our understanding of anisogamy and oogamy.

Thank you. According to your and editor's suggestion, we have deleted the discussion and Figure 9 about the evolution of anisogamy and oogamy from the manuscript and limited the scope of the manuscript to focus on algal biology and the role of MID in MSP formation.

2. The authors persist in using language that describes beta MSP as dominant eg. line 18 "type β is the dominant phenotype," and lines 84/85, 148, 159. As was pointed out previously, there is no dominance relationship between MSPs. MSPs are under the control of MID or mating type determining genes, and it is these mating type determining genes that display a dominance relationship, not the MSPs.

According to your suggestion, we have replaced "dominant phenotype" with other expression which were highlighted within the document by purple color text.

3. The authors make a point about evolution of MSPs and their relationship to the plus and minus mating types in *Chlamydomonas* versus *Ulva* (lines 355-357). Unfortunately, this is an empty argument as mating type assignment for isogamous algae was previously decided based on somewhat arbitrary criteria. In the case of *Chlamydomonas* it was uniparental chloroplast inheritance that was designated as the maternal parent (mt+), but if mitochondrial inheritance had been used as the criterion then the assignment of mating types would have been reversed. It is interesting that in both species the dominant mating type is associated with the beta-type MSP, but with a sample size of two species there is not much to conclude about this observation.

*Thank you for your valuable suggestion. As you pointed out, mating type assignment for isogamous algae is based on arbitrary criteria. Therefore, we deleted the discussion about evolution of MSPs and their relationship to the plus and minus mating types in *Chlamydomonas* versus *Ulva*, and we also deleted examples of species from the Figure 8 (Fig. 6 in revised manuscript).*

*Instead of such discussion, we discussed the genetic control of MSPs in *Ulva* and other ulvophycean species which is assumed based on the results of present study (lines 256-265 in revised manuscript). We also discussed the relationships of MSPs with other mating type (sex) specific traits (e. g. uniparental inheritance of organelles) among *C. reinhardtii*, *Ulva* and other ulvophycean algae (lines 265-272). Moreover, we*

mentioned the interesting association of dominant mating type with type beta phenotype which is commonly found in two distantly related species, C. reinhardtii and U. prolifera (lines 272-279). We also mentioned the sample size is not large enough to conclude the significance of this phenomenon (lines 279-280).

4. The additional experiments that are meant to test the impact of eyespot placement in planozygotes are flawed and not convincing. The underlying defects of mlt (multi eyespot) mutants are not well understood. While mlt strains have extra eyespots, they may also have cytoskeletal defects or other defects that impact phototaxis independently of the extra eye spots.

Thank you for your valuable comment. As you pointed out, it is difficult to rule out the possibilities that cytoskeletal defects or other defects impact phototaxis of mlt mutants. Therefore, we deleted the results of phototaxis experiments and Figures.

5. I don't think that demonstrating phototaxis defects in planozygotes with mismatched eye spots (i.e., those that fuse with eyespots on opposite sides) is necessary. The ability to form asymmetrically placed mating structures in gametes of opposite mating type is conserved, and to me that is a strong argument for MSP dimorphism being under selection. Accordingly, the long speculative paragraph starting on line 297 could be significantly shortened. The problem with testing phototaxis is that even if it could be done convincingly, it is possible that the asymmetric MSP configuration is important for something else, such as proper organization of the meiotic cytoskeleton, so proving/disproving its utility for phototaxis would not rule out other roles where there might be even stronger selection.

Thank you for your valuable suggestion. According to your suggestion, we deleted the results of phototaxis experiments and shortened the paragraph starting on line 297 (line 239 in revised manuscript). In this paragraph, we briefly discussed that MSP configuration in gametes, coordinated alignment of flagella and eyespot(s) and phototactic behaviour of planozygotes are conserved in chlorophytes, and that sufficient selective pressure are acting on these traits. We also mentioned the possibility that gamete MSP is important for other cellular functions.

Minor Points

6. Please use "sexes" and "mating types" appropriately. Chlamydomonas and other isogamous algae do not have two sexes, they have two mating types.

Thank you. We have tried to use “sexes” and “mating types” appropriately in the revised manuscript which were highlighted within the document by red color text.

7. Lines 13/14. “based on the sex-specific asymmetric positioning of the mating structure”. Asymmetric position with respect to what?

Thank you. We added “with respect to the flagellar beat plane and eyespot” to the sentence (lines 14-15 in revised manuscript).

8. Line 24. Eukaryotes do not generally have two sexes or mating types. While two mating types are most common, there are many examples of multiple mating types. Multiple sexes are not observed. Even with hermaphrodites there are still only two sex roles.

According to your suggestion, we used “sexes” and “mating types” appropriately in the revised manuscript. In this sentence, we rewrote as follows, “Eukaryotes generally have two sexes....., and two or more mating types in isogamy.”(lines 24-25 in revised manuscript)

9. Lines 82-84. This is an awkward sentence that needs a rewrite, perhaps splitting into two sentences. Sex-reversed is inaccurate language. The strains are mating-type reversed.

Thank you. We rewrote this sentence as follows, “We observed the positioning of the mating structure of the wild-type and mating type reversed strains.....(FE-SEM). We demonstrated that the spatial positioning of the gamete mating structure was....by MID”(lines 84-89 in revised manuscript).

10. Lines 333-6. Unclear sentence. What were the mating specific genes used for? PCR genotyping?

Thank you. We rewrote this sentence as follows, “, which were identified using genome sequencing of both mating types of U. partita and confirmed as mating type specific gene by PCR based genotyping in U. prolifera”.(lines 259-261 in revised manuscript)

REVIEWERS' COMMENTS:

Reviewer #5 (Remarks to the Author):

I am glad to see the authors removed the off topic parts of their manuscript which I feel is much better in its streamlined form.

I still dislike the first sentence which should be rewritten. Most eukaryotes are single celled protists and do not have sexes, so the first sentence starts with a misleading statement. Two sexes are common in multicellular eukaryotes, but isogamy is probably the norm.

Dear Reviewers

Thank you for your valuable comments and suggestions. We have studied the comments of reviewer#5 carefully, made the necessary corrections and rewrote the manuscript according to the reviewer's suggestions, which are highlighted within the document by red color text.